# Mechanisms of cultural diversity in urban populations

Harin Lee [1,2,3,4] ✉, Nori Jacoby [3,5], Romain Hennequin [1,6] & Manuel Moussallam [1,6]

Large cities exhibit greater cultural diversity. Due to limited data on individual behaviour, previous research could not discern whether this stems from demographic heterogeneity or enhanced individual cultural exploration. Analysing 250 million listening events from 2.5 million users across France, Brazil, and Germany, we investigate mechanisms driving urban cultural diversity. We assess the collective shared musical repertoire in each geographical area, while concurrently measuring individuals' music engagement breadth through listening histories. Cross-culturally, both collective diversity and individual breadth increase with population size, aligning with cultural evolution and urban scaling theories. While demographic factors such as age, gender, income, immigration, education, and social connections influence these trends, especially in highly populated areas, they do not fully explain the observed patterns. This suggests large cities are culturally diverse not only because they aggregate people from varied backgrounds but also due to greater opportunities created for cultural interactions and discovery.

Urban life is vastly different from the experience of smaller cities and towns. Inhabitants of large metropolitan areas tend to live a faster paced lifestyle[1,2], engage in more social interactions[3], and the limited physical spaces allow for more rapid dissemination of information but also of diseases[4,5]. Recent census data from various countries reveal a consistent pattern. As cities grow in size, their collective outputs in various aspects, such as economy and education, surpass linear growth expectations[6–9], a phenomenon known as the urban scaling law. Cultural richness and complexity also expand with population size. Computer simulations[10–15] and experimental studies[16–19] demonstrate that larger populations are not only effective at preserving cultural complexity and diversity but can also stimulate innovation through increased cultural exchanges. Despite these insights, the exact mechanisms and factors contributing to cultural diversity remain a topic of debate[20–23].

Of all the possible mechanisms, two hypotheses offer concrete predictions. The 'demographic mixing' hypothesis suggests that cultural diversity in densely populated cities is a result of demographic variances[1,24–26], attracting individuals from diverse backgrounds, as

seen in megacities like Paris. In contrast, the other hypothesis rather focuses on the wider 'cultural breadth' of individuals living in those cities, proposing that continuous exposure to diverse and rich cultural inputs impacts one's psychology and broadens their cultural taste[27–30], leading to a communal shift towards greater diversity. These two mechanisms could also be interconnected. For instance, demographic mixing could create opportunities for interaction and exchange with individuals from more diverse cultural backgrounds, in turn, serving as a catalyst for new discoveries outside one's usual consumption. Alternatively, sheer size alone could independently drive cultural diversity, as larger populations are more effective at preserving cumulative cultural knowledge[11,31].

Previous studies have struggled to distinguish between these hypotheses, primarily due to the reliance on aggregated population-level data that overlooks individual behavioural patterns and demographics. Furthermore, while the link between socio-economic status and cultural breadth is well-studied (e.g., cultural omnivory[32]), there is a lack of empirical evidence on how the richness and size of one's

[1]Deezer Research, Paris, France. [2]Max Planck Institute for Human Cognitive and Brain Sciences, Leipzig, Germany. [3]Max Planck Institute for Empirical Aesthetics, Frankfurt am Main, Germany. [4]Department of Life Sciences, Leipzig University, Leipzig, Germany. [5]Department of Psychology, Cornell University, Ithaca, NY, USA. [6]These authors contributed equally: Romain Hennequin, Manuel Moussallam. ✉e-mail: hlee@cbs.mpg.de

cultural environment might shape their cultural taste. To bridge these gaps, we analyse a large dataset of music-listening events in the real world. Music serves as an ideal test-bed for examining questions about cultural diversity, with its universal prevalence[33,34] and cultural distinctiveness[35–37], and significance in people's everyday lives[38,39]. Prior research, such as a study on Irish folk music by Street et al.[40], showed that tunes played by a larger community of musicians evolve into more varied versions and exhibit intermediate levels of melodic complexity.

Including the work by Street et al., most research on music diversity has focused on the producer's perspective—analysing the varying types of musical artefacts that are produced[33,34,41,42]. However, both production and consumption, and their feedback loops with underlying selection pressures, are crucial for a comprehensive understanding of cultural evolutionary processes[43], particularly in the modern world of rapidly diffusing information and trend dynamics. In the current study, we focus on the consumer's perspective—analysing the varying ways in which individuals engage and listen to music[44,45].

Over four weeks, we track over 250 million listening events across 2,544,548 users on Deezer, a global music streaming platform available in 187 countries featuring over 90 million tracks. We collect data from France, Brazil, and Germany, which are the three countries with the largest percentage of Deezer users relative to their general populations. Using municipalities and NUTS unit geographic boundaries, we assess population-wide diversity in listening patterns by examining the alignment of music listening between individuals. Concurrently, we examine these individuals' breadth of music engagement by tracing their listening histories. This dual perspective of diversity allows us to decipher different ways in which size affects cultural diversity outcomes both between and within individuals. The cross-cultural aspect of the data allows us to test the generalisability of the phenomenon.

To assess the extent to which the level of diversity might arise from the heterogeneity of the population (i.e., demographic mixing) or the broader spectrum of cultural engagement (i.e., cultural breadth), we account for various socio-demographic factors using simple linear regressions and causal inference tools. The demographic factors include self-reported age and gender, as well as inferred levels of income, education, immigrant status, access to music venues, and social connections. The inferred data is derived from various sources, offering detailed granularity at the level of over 35,000 municipalities, equivalent to postcode area resolution. We achieve this by pinpointing each user's home location and matching it to the corresponding demographic information from these sources. This expansive, spatially and temporally detailed dataset (openly available at https://github.com/harin-git/mus-div) offers a unique opportunity to understand how local population size and demographic nuances influence multiple layers of cultural diversity.

## Results

Figure 1a shows the user distribution in our data. The data spanned over four weeks in March 2023 across users in France, Brazil, and Germany. After filtering for users only with reliable geolocations (User sampling in Methods), the final dataset encompassed over 2.5 million users (Fig. 1a) across France ($N = 1,506,899$), Brazil ($N = 816,101$), and Germany ($N = 221,548$). Detailed demographics and comparison with census data can be seen in the 'User demographic' section in Methods. To ensure a balanced contribution from each user, we drew a random 100 unique streams per user. Users were grouped according to the geographical boundaries of area units (User sampling in Methods), and we discarded areas with less than 200 unique users to deal with noisy data, and to ensure further anonymisation at the aggregate level. We analysed the data after anonymisation and the computations were not used to derive any commercial user profiling of any kind. All analysis scripts used in the study and aggregated anonymised data are openly available (Data availability).

### More distinct music interests in large metropolitan areas

We begin by assessing the diversity of music listening patterns between individuals, analysing how similar or dissimilar one's listening is to another residing in the same area. This is measured through between-individual diversity (BID), which adopt Hill's number (with order of 1, which effectively becomes the Shannon entropy; Measuring diversity in Methods), a widely used method for measuring cultural diversity that allows for standardised comparisons[46–48]. A high overlap of songs among individuals indicates a more shared consumption of musical repertoire, resulting in a greater skewness in the frequency distribution, where consumption is concentrated on a highly popular set of songs (i.e., low diversity, low BID). Conversely, a low overlap indicates there being a greater distinction between individuals, resulting in a flatter frequency distribution, where consumption is more widely distributed (i.e., high diversity, high BID).

To ensure that varying user sample sizes across areas do not bias our diversity estimates, this analysis and all statistics reported in the paper are standardised by drawing an equal number of samples at every level of analysis and computing bootstrap estimates Measuring diversity and Statistical analysis in Methods).

Using geographical boundaries of NUTS3 unit areas of France and Germany, and municipalities of Brazil (User demographic in Methods), we assess the level of BID relative to the population size of the area (derived from the number of users in the given area, log-transformed with base 10). In all three countries, there were strong positive correlations (Fig. 1b; France: Two-tailed Pearson $r = 0.73$, 95% CI = [0.62, 0.40], $P < 0.001$; Brazil: $r = 0.40$ [0.28, 0.51], $P < 0.001$; Germany: $r = 0.79$ [0.70, 0.85], $P < 0.001$; see Supplementary Tables 1,2 for additional correlations including algorithmically recommended streams and using Spearman method). This indicates that individuals living in more populated areas tend to have more distinct listening patterns from one another. Interestingly, our analysis also revealed notable between-country differences (Fig. 1b inset; one-way ANOVA: $F_{(2, 424)} = 371$, $P < 0.001$). BID in Germany (Mean aggregate across areas = 962 [960, 962]) was substantially higher than in France (Mean = 936 [934, 937]; Germany vs. France: Cohen's $d = 3.04$, $P < 0.001$), and Brazil (Mean = 886 [882, 891]; Germany vs. Brazil: Cohen's $d = 2.69$, $P < 0.001$). Together, these results suggest a country-specific characteristic in the amount of shared music repertoires, alongside a country-independent, universally positive trend between BID and local population size.

To examine the robustness of these results, we conduct a series of control analyses. First, we expand our groupings of analysis and test beyond frequencies of 'songs', to 'artists' and 'musical genres' (Fig. 1c) and found these positive correlations to hold (grouping by artist: $rs > 0.40$; grouping by genre: $rs > 0.33$; all $Ps < 0.001$; Supplementary Table 1). Second, we account for various biases that might stem from our user filtering criteria (Supplementary Fig. 2). Third, we extensively test for regional differences in the usage of algorithmic recommendations and their influence on the observed outcome (Supplementary Tables 2 and Supplementary Fig. 3; Algorithmic streams in Methods). Fourth, we explore alternative population size metrics, using census population and density (Supplementary Fig. 4), and employ alternative diversity metrics with varying parameters of Hill's order and using the Gini-coefficient (Supplementary Table 3). Finally, we examine the extent to which spatial autocorrelation is present and test for spatial effects through detrending analysis and broader geographical groupings (Supplementary Table 4). We observed consistent patterns across all of these cases: BID and local population size were highly correlated even when these confounders were tested empirically.

Thus far, all our analyses have been based on a single approach measuring BID. For further validation, we adopt an alternative approach for measuring BID, recognising that measuring diversity purely by frequency counts might overlook the relationships between the songs. We leverage a high-dimensional vector space that

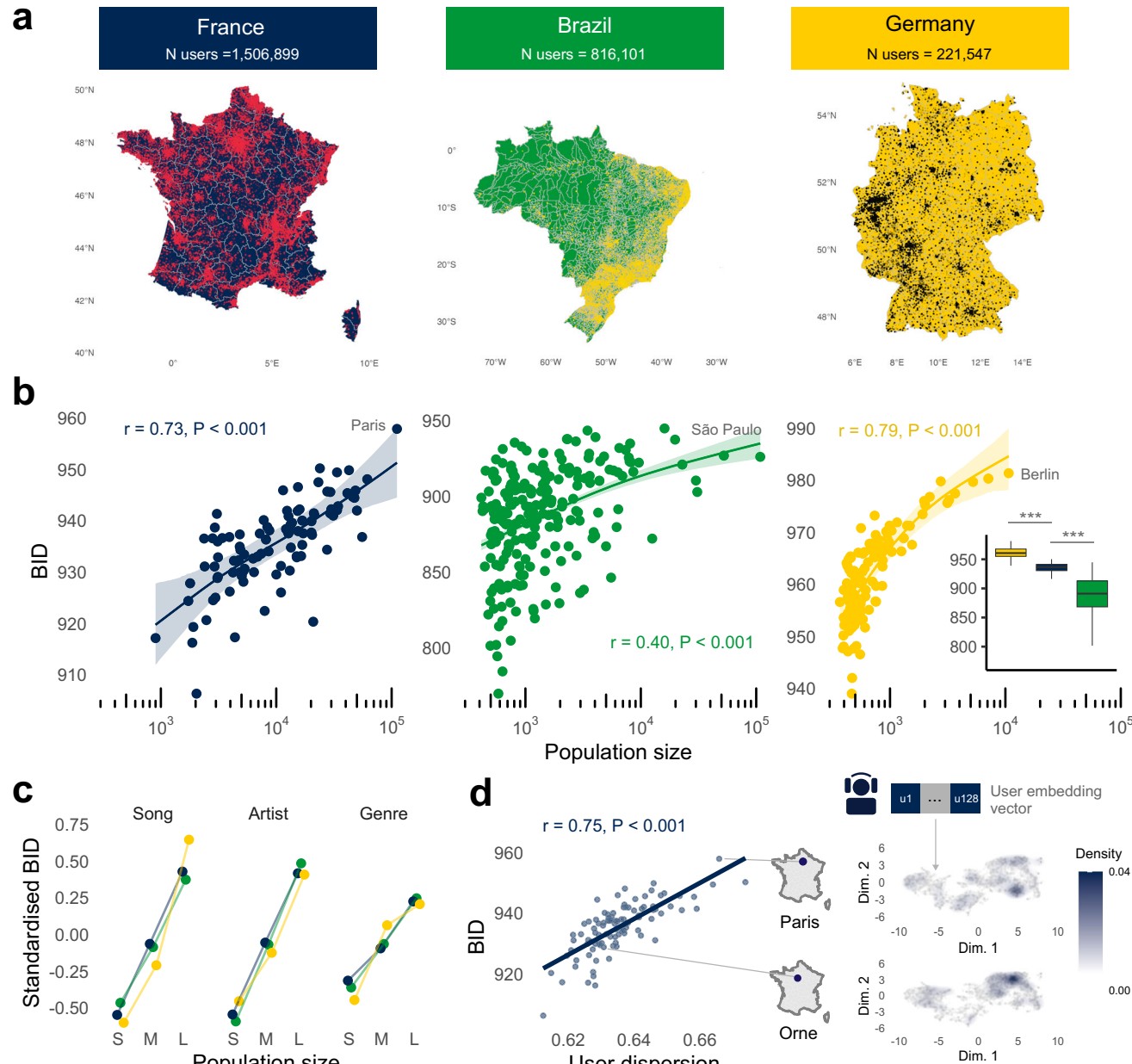

**Fig. 1 | Individuals' music listening is more distinct from one another in large metropolitan areas, resulting in dispersed pockets of diverse music interests.** **a** Geolocations of users sampled from France ($N$ = 1,506,899), Brazil ($N$ = 816,101), and Germany ($N$ = 221,547). **b** Between-individual diversity (BID) is measured using Hill's number and computed across the geographical boundaries of NUTS3 unit areas of France ($N$ = 96) and Germany ($N$ = 113), and municipalities in Brazil ($N$ = 218; Measuring diversity in Methods). Each dot represents the mean estimate from sampling 1000 unique music streams across bootstrap simulations (Statistical analysis in Methods; see Supplementary Table 2 for additional analyses including algorithmically recommended streams). The resulting BID value X indicates that the diversity is equivalent to having X cultural items (i.e., songs, artists, genre categories) that are all equally represented in the dataset. Curved lines are generalised additive model (GAM) fitting between BID and log base 10 of population size. Shaded areas and error bars represent 95% CI of the bootstrap mean. Two-tailed Pearson correlation is used and corrected for multiple comparisons. Inset box and whiskers (median, 25% and 75% quantile hinges, 1.5x interquartile range whiskers) show between-country differences with two-tailed post-hoc Tukey

indicating significance levels at $P < 0.001$ (***), $P < 0.010$ (**), and $P < 0.050$ (*). **c** Alternative units of analysis, grouping streams by 'artist' and 'musical genre', with standardised units and three quantile population size categories (see Supplementary Table 1 for raw correlation values and using Spearman method). **d** Two-tailed Pearson correlations between BID and user dispersion in the high-dimensional vector space. User dispersion is measured by first defining each user as a 128-dimensional vector representation, and then performing dimensionality reduction using UMAP to construct the user embedding space (User embedding in Methods). Users that are closer in this space tend to have more similar music interests. Two areas are selected as examples to demonstrate the dispersion of users using kernel density estimation (Dispersion in user embedding in Methods; see Supplementary Fig. 5 for more density maps). More dispersed density is observed across Parisians, indicating they have more distinct listening patterns from one another. In contrast, users from Orne, a small area with less than 300,000 inhabitants, demonstrate a more concentrated hotspot, suggesting a more homogenous taste among its residents.

summarises the *relations between users* based on the similarity of their listening behaviour, where users who listen to similar music are situated closer in this space (e.g., both are K-pop fans), while far apart if they have distinguished interests (e.g., K-pop versus metal fans; User embedding in Methods). The amount of variance in the pairwise distance among the users is then used to capture the general homogeneity or heterogeneity in music interests of the local area (Dispersion in user embedding in Methods).

Using France as a case example, Fig. 1d demonstrates the strong correlation between the frequency-based BID measure and the amount of dispersion of users in the high-dimensional vector space ($r = 0.75$ [0.65, 0.83], $P < 0.001$). The dispersion amount also correlated highly with population size ($r = 0.54$ [0.38, 0.67], $P < 0.001$), validating the robustness of our results. We include two areas as insets in Fig. 1d, as illustrative examples, to demonstrate the concentration of users in the vector space using kernel density estimation (see Supplementary Fig. 5 for additional examples). The two density distributions demonstrate that the Parisian listeners are more scattered and dispersed across a broader space, indicative of there being pockets of diverse music interests. Conversely, the listeners from Orne (a smaller area in the west of France) are more concentrated within a narrower space, demonstrating that they have more shared, homogeneous tastes.

To summarise, our analyses consistently reveal significant disparities in listening patterns between individuals residing in large metropolitan areas and those in rural areas.

## Wider personal breadth of music engagement in large metropolitan areas

While previous studies were limited to measuring diversity at the population scale (equivalent to our assessment of BID), we further leverage the individual-level data to understand how an individual's breadth of music engagement may be associated with the size of the environment in which they reside.

This is measured through within-individual diversity (WID), where we adopt a metric known as the Generalist-Specialist Score (GS-Score) of music engagement[45,49] by tracing each individual's listening history (Measuring diversity in Methods). Here, we again utilise a high-dimensional vector space, but this time, capturing the *relationships between songs*, created based on millions of user interactions with songs on the Deezer platform (Song embedding in Methods). Next, we calculate each user's 'coverage' in this vector space by mapping all the songs they have listened to in the past 28 days. This coverage, or radius in the vector space, is used as an indicator of one's overall breadth of music engagement. If a user is a 'specialist', their listening habits will be concentrated in a specific location of the vector space (for example, they may only listen to K-pop), and their radius will be small. Conversely, a 'generalist' user will have a wider coverage, indicating they listen to a diverse range of music (for example, they may listen to K-pop, metal, and classical music). To compute this radius, we use the average cosine similarity between a randomly selected song and the average of all songs the user has listened to (Measuring diversity in Methods).

Using the same sampled users and geographical units, we assess the level of WID relative to the population size of the area. Mirroring the results of BID, we observed similar, albeit weaker, positive trends across all three countries (Fig. 2a; France: Two-tailed Pearson $r = 0.65$ [0.52, 0.75], $P < 0.001$; Brazil: $r = 0.32$ [0.20, 0.44], $P < 0.001$; Germany: $r = 0.34$ [0.17, 0.49], $P < 0.001$; see Supplementary Fig. 6 for correlation between WID and BID).

While we previously saw substantial between-country differences in the level of BID, the three countries exhibited similar levels of WID (Fig. 2a inset). Brazil (Mean = 39.5 [39.3, 39.7]) had the highest level of WID compared to Germany (Mean = 39.1 [38.9, 39.4]; Brazil vs. Germany: Cohen's $d = 0.30$, $P = 0.010$) and France (Mean = 39.0 [38.7, 39.2]; Brazil vs France: Cohen's $d = 0.43$, $P < 0.001$), but the effects

were much smaller in size, and there was no significant difference between Germany and France ($P > 0.05$). These results suggest that individuals who reside in more populated areas tend to engage with a broader spectrum of music content, potentially stimulated by more diverse inputs coming from the richer cultural environment. At the same time, the small differences between countries in the general size of this breadth indicate that an individual's cultural background does not play a substantial role in shaping their breadth.

Among the many factors that could contribute to one's breadth of engagement, 'age' has been noted as a strong predictor of consumption diversity in previous experimental research[50,51] and observational data of online behaviour[44]—suggesting that people tend to explore and discover more during adolescence while their preferences become more stable with age. However, to our knowledge, no study to date has empirically demonstrated this trend in music consumption with detailed annual granularity across the lifespan.

As a validation of our method for measuring within-individual diversity and to extend previous findings, we analyse WID as a function of age. Figure 2b demonstrates a remarkably consistent, inverted U-shape pattern across all three countries. This cross-cultural pattern shows that individuals' musical experiences broaden from their teenage years into their early 20s, reach a peak in their late 20s, and then gradually narrow as they get older.

Notably, this trend persisted even when accounting for variations in baseline streaming activities across different age groups—as younger individuals tend to stream more music (Supplementary Fig. 7). We also replicated the inverted U trend using an alternative approach by measuring one's diversity through their varied listening across musical genres (Supplementary Fig. 8). This validation was crucial considering that relying on song embeddings to compute WID can inherently be biased due to the skewed demography of Deezer users towards younger generations (Supplementary Fig. 9).

It is important to note, however, that computed WID here reflects both the real age of the user as well as the period they were born in. Therefore, a cohort effect may exist whereby individuals born in a particular era have more or less diverse musical tastes. Disentangling these associations would require longitudinal data that tracks individuals' change in music taste diversity over an extensive period of time. Future studies can incorporate datasets that track individuals' listening behaviour over many years[52].

## Diversity is largely explained by demographic factors, but not entirely

The strong influence of age on WID exemplifies the necessity to control for socio-demographic confounders that can bias our estimates of the relationships between the two levels of diversity and population size. Larger cities tend to attract younger, more educated, and wealthier individuals who may demonstrate 'cultural omnivore' tendencies, embracing broader musical preferences from elite to popular genres[30,53–55]. Metropolitan areas are often also international hubs where diverse cultural activities concentrate, such as there being more music events. Such unique properties of large cities are in stark contrast in their demographics compared to their rural counterparts. In fact, several past studies have found that the influence of population size on diversity to reduce, or even diminishes, when controlling for other demographic differences, migration, and social connections[14,54,56].

To address the question of whether the increased diversity we observe is a result of demographic differences between urban and rural areas, we go beyond correlational analyses. We focus on France, given that our user sample has the highest spatial granularity there and the greatest availability of dense, high-quality demographic data. We collect comprehensive data on various socio-demographic attributes at the level of over 35,000 municipalities and 96 NUTS3 unit areas (see Supplementary Fig. 10 for demographic disparities by area size). First,

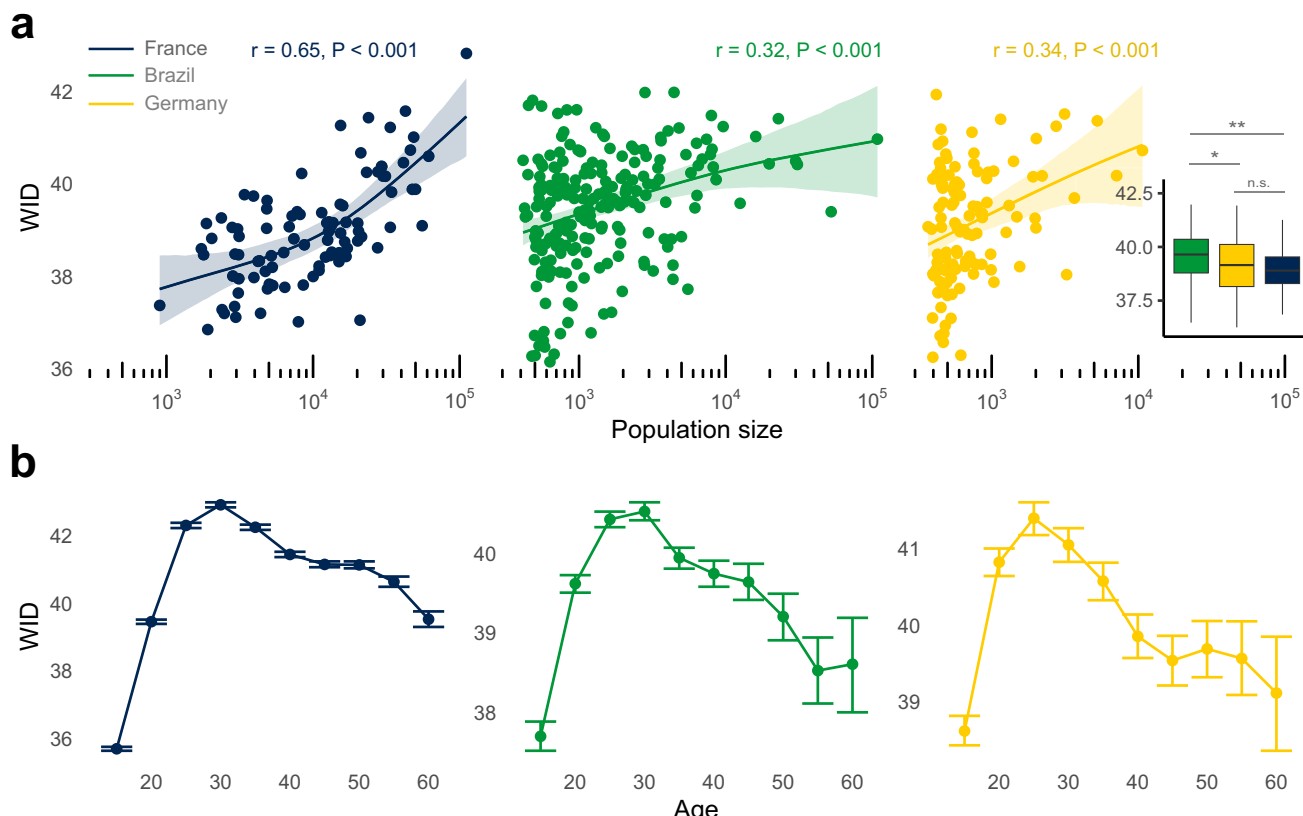

**Fig. 2 | Individuals living in large metropolitan areas engage with more diverse music. An individual's breadth of engagement follows an inverted U-shape trajectory over the course of life. a** Relationship between population size (N unit areas: France = 96, Germany = 113, Brazil = 218) and within-individual diversity (WID) with values ranging from 0 (least diverse) to 100 (most diverse). The metric captures the coverage of one's musical engagement within the high-dimensional vector representation of songs (Measuring diversity in Methods; see Supplementary Table 1 for correlations using Spearman method). Curved lines are generalised additive model (GAM) fitting between WID and log base 10 of population size. Shaded areas and error bars represent 95% CI of the bootstrap mean. Two-tailed

Pearson correlation is used and corrected for multiple comparisons. Inset box and whiskers (median, 25% and 75% quantile hinges, 1.5x interquartile range whiskers) show between-country differences with two-tailed post-hoc Tukey indicating significance levels at $P < 0.001$ (***), $P < 0.010$ (**), and $P < 0.050$ (*). **b** WID as a function of age. Cross-culturally, WID forms an inverted U-shape trajectory with the peak of most diverse exploration around the late 20 s. Each dot shows the bootstrapped mean estimate of 100 randomly selected unique individuals across the 5-year age group bins. Error bars represent 95% CI of the bootstrapped mean (Statistical analysis in Methods).

we gather commune level census data from Eurostat and the National Institute of Statistics and Economic Studies (INSEE), covering median income, proportion of immigrants, and proportion of university degree holders (see Supplementary Fig. 11 for map visualisations). Second, we approximate the accessibility to music events in each area by mapping the locations of 6618 music venues across the country using the SongKick database—a popular website for tracking music events (see Supplementary Fig. 12 for map visualisation). Third, we include data from Facebook to approximate the number of international social connections (see Supplementary Table 6 for demographic comparison to Deezer users). Fourth, we consider differences in the usage of algorithmic recommendations across the areas (Supplementary Fig. 3). Lastly, we obtain self-reported age and gender from Deezer's registration information. All details regarding the justification of considering these demographic variables as potential confounders, along with the data source and the collection method, are described in the 'Socio-demographics' section in Methods.

We noticed significant correlations between demographic variables and diversity, as indicated by the correlation matrix (Supplementary Fig. 13). To further investigate this relationship, we perform simple linear regressions to assess the extent to which demographic factors account for the diversity. We first categorise user samples into three population quantile categories with each containing 32 areas: small (census inhabitant Median = 228,750), medium (Median =

542,122), and large (Median = 1,260,378). Next, for each population size category, we separately regress BID and WID on all demographic factors. The adjusted R-square values showed that demographic variables explain 28.2% of BID and 36.6% of WID in small areas, 53.9% of BID and 41.5% of WID in medium areas, and 80.1% of BID and 84.1% of WID in large areas. This suggests that high levels of BID and WID in the largest metropolises mainly arise from their demography.

Nevertheless, such a simple model ignores the interaction among the variables, and these interactions can bias the observed estimates in various ways (see Cinelli, Forney & Pearl[57] on 'bad controls'). We therefore introduce a more complex model by making assumptions about the relationships among these variables based on the existing literature (Socio-demographics in Methods). We illustrate this in the form of a Directed Acyclic Graph (DAG). DAG provides a clear and efficient method to identify, present, and hypothesise the causal relationships between variables[58,59]. Importantly, DAG allows the identification of which confounder should be controlled or left uncontrolled, rather than controlling for every imaginable covariate. Using this DAG, we test the direct effect of population size on BID (Fig. 3a) and WID (Fig. 3b) after controlling for demographic variables. We checked that several key assumptions are met, including implied conditional independence tests to ensure that the graph we assume is not refuted by the observed data (Causal inference in Methods). The step-by-step procedure we applied for causal testing is detailed in

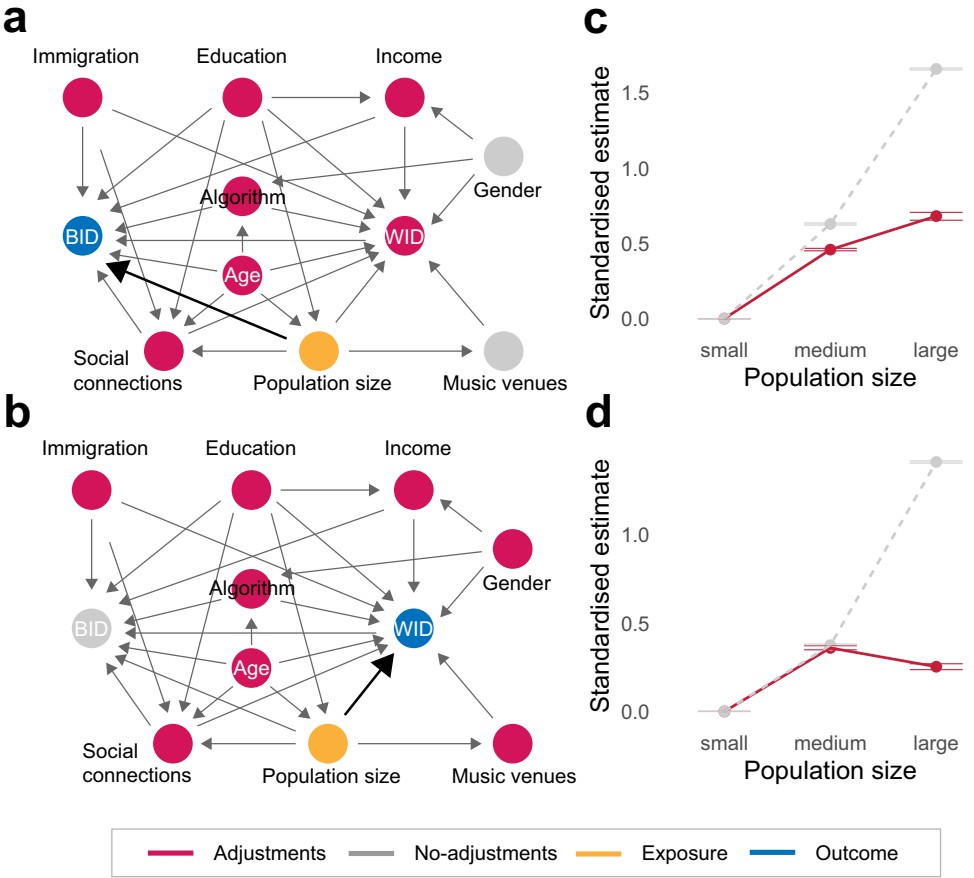

**Fig. 3 | Demographic variables largely account for the diversity in the most populous areas and reveal distinctions between BID and WID.** DAG models for testing the direct effects of population size (orange) on (**a**) BID and (**b**) WID (blue). Variables are aggregated at the NUTS3 unit granularity, resulting in 96 area data points. Adjusted variables in the models (i.e., minimal adjustment set) are coloured in red, whereas unadjusted variables are coloured in grey (Causal inference in Method). Bold arrows indicate the causal paths of interest. **c**, **d** To control for the confounders, inverse propensity weights are applied to users categorised into three quantile population groups (32 geographical areas in each category). Across these population size groups, model outcomes on (**c**) BID and (**d**) WID are plotted, comparing adjusted (solid, red) and unadjusted (dashed, grey) effects. The estimates are standardised using Z-scores with intercepts matched at the 'small' population group for effect size comparisons. Each dot represents a bootstrapped mean estimate with error bars indicating 95% CI from the bootstrap mean (Statistical analysis in Methods).

Supplementary Note 1, which generally adopts the protocols proposed by Ankan et al.[60].

In the two models we test, first we identify the confounders (red in Fig. 3a, b; i.e., minimal adjustment sets[57]) and adjust them to obtain a direct estimate of the effect of exposure (orange in Fig. 3a, b) on the outcome (blue in Fig. 3a, b). Next, we assign inverse propensity weights (IPW) to each user based on the average treatment overlap (ATO) population method[61,62]. This procedure balances the distribution of the confounding variables across the groups (see Supplementary Note 1 for details). IPW is a widely used modern causal inference technique for creating an artificial scenario that mimics a randomised trial (for a review on causal inference methods, see Chatton & Rohrer, 2023[62]). The standardised mean difference (SMD) tests (Supplementary Fig. 14) and the empirical cumulative distribution function (ECDF) of the weights (Supplementary Fig. 15) indicated that the groups are well balanced after applying the weights. Lastly, we ran bootstrap simulations of the models by sampling users and assigning new weights in each iteration to obtain confidence estimates (Statistical analysis in Methods).

Figure 3c, d show the results contrasting the unadjusted effects (dashed and grey lines) with those adjusted using IPW (red and solid lines). The variables are standardised using Z-scores and the intercepts are matched at the 'small' population group for effect size comparisons (see Supplementary Fig. 16 for trends stratified by each

confounder separately). After adjusting for demographic differences, the general trend remained: medium and large areas exhibited greater cultural diversity than smaller ones ($P$s <0.001). However, it also revealed notable nuances, especially in the largest population quantiles. Consistent with the simple regression analysis reported earlier, demographic factors accounted for much of the cultural diversity in the largest areas. However, this was more pronounced for WID than BID (reduction in standardised estimate: WID = 1.16 [1.14, 1.18], BID = 0.98 [0.95, 1.01]). As a result, BID remained a consistent linear increase with population size (standardised estimate from intercept: medium = 0.46 [0.45, 0.47], large = 0.68 [0.65, 0.71]), whereas WID levelled off after the medium quantile (medium = 0.36 [0.35, 0.37], large = 0.25 [0.24, 0.27]).

To conclude, the findings indicate that while demographic mixing contributes significantly to the cultural diversity of highly populated areas (e.g., capitals), it does not entirely account for the ongoing increasing trend in diversity between individuals (Fig. 3c). Conversely, the plateau observed in WID beyond a certain population threshold (around 500,000 inhabitants) implies that there is a saturation point for individuals' cultural exploration (Fig. 3d).

## Discussion

Prior observations of heightened cultural diversity in large populations were limited in understanding whether this increase is driven by the

population's demographic heterogeneity or by broader cultural engagements of individuals. Using music listening patterns in the real world, we sought to decipher this and study the mechanisms and factors contributing to cultural diversity, here defined as diversity in music consumption. We distinguished different layers of diversity by simultaneously measuring (1) the shared listening patterns between individuals, and (2) within-individual breadth of music engagement.

Consistent with previous findings on the association between a population's cultural diversity and its size[10,11], we observed an increasing trend of between-individual diversity with the population size of the area. We add new insights using individual patterns: an individual's cultural breadth similarly expands with population size. This provides supporting evidence for the 'cultural breadth' hypothesis that has thus far remained largely theoretical. These patterns were consistently observed across France, Brazil, and Germany. However, across these countries, we also found noticeable group-level differences for between- but not within-individual diversity. This suggests that people from different countries are generally similar in their cultural breadth, but nations differ in the amount of shared repertoire among those people. We confirmed the robustness of our results through a series of control analyses that account for various biases that might stem from the data or the measure. These insights have practical implications such as guiding policymakers to distribute cultural resources across regions effectively[63].

We demonstrated how demographic factors can contribute to these observed trends. In particular, we showed a consistent cross-cultural pattern where an individual's breadth of cultural exploration tends to peak in their late 20s and narrows as they get older. This finding augments prior research that established a connection between age and diversity in cultural consumption[44,51], by providing a more detailed perspective through leveraging large-scale data that is otherwise challenging to obtain.

Using causal inference tools, we found that demographic factors largely contribute to diversity in the most populated areas. However, they did not fully account for the observed trend of increasing diversity with population size. This suggests that cultural diversity in large metropolises is not only due to demographic mixing but also results from additional mechanisms. One such mechanism may be the more frequent cultural interactions and exchanges that larger populations enable, promoting diversity through increased interpersonal contact. Another is the enhanced global connectivity of big cities, as international hubs, they facilitate the introduction and integration of global cultural products into local traditions. Interestingly, once demographic factors are accounted for, there was no discernible difference in the cultural breadth of individuals living in medium (inhabitant size around 500,000) versus large areas (more than 1.2 million), suggesting a possible upper limit in the extent that size fosters cultural exploration. This limit could potentially arise from the availability of sufficient cultural infrastructures to support diverse cultural experiences once cities become sizeable enough, though this conclusion may require further replication in other countries to account for potential confounds resulting from other demographic factors.

Our consumer-centric perspective on cultural diversity complements the producer-centric perspective that has been the dominant form in studies focusing on population size and culture[11,31,64]. Previous studies of musical diversity have primarily focused on the content itself, such as the melody and rhythm of songs[33,34,40], considering these as the units of cultural traits. Our study, however, fills an important gap by analysing an often overlooked dimension: individuals' musical tastes as cultural traits that can evolve and are transmittable. Importantly, cultural variations within a population can take multiple forms, including differences in production[11], preference or consumption[32,44], and perception[37]. A comprehensive understanding of cultural diversity necessitates considering all these aspects.

Cultural traits resulting from production and consumption also interact intimately through feedback cycles, which are common in complex systems, including culture and economy[43,65,66]. For instance, a diverse range of consumption tastes might stimulate producers to create a broader array of cultural products, with social feedback signalling this demand. Conversely, individuals exposed to a wider range of cultural inputs, and consequently possessing a broader cultural breadth, may tend to develop cultural niches. Furthermore, consumption patterns may not only influence the range of what is produced but can also inspire innovations through blending of genres, or the emergence of entirely new forms. When consumers show interest in more diverse cultural products, producers may respond by experimenting with new combinations of styles and ideas to meet that demand. This increased diversity in production patterns could then further enhance diversity in consumption, reinforcing the feedback loop. Hence, understanding listening behaviour can explicitly characterise how social selection pressure biases the creation of music. Moreover, selection biases significantly influence the cultural production studied from historical records, as artefacts that were widely consumed are likely to have survived and thus been documented[67]. The dynamic interplay between production and consumption could reveal deeper insights into these co-evolutionary processes. Future studies might consider simultaneously measuring the diversity of cultural products and consumption, relating them to population size and demography.

Our findings also suggest the presence of other potential evolutionary mechanisms. Unlike genetic sequences, which primarily rely on vertical transmission (i.e., parents passing their genes to offspring), cultural evolutionary processes are thought to exhibit significant variations within the same generation (horizontal transmission) through social learning[68]—music sharing among friends for instance. Notably, an individual's cultural trait is not static like genes. One's musical taste and their breadth of exploration can change even within their lifespan, as seen with our results of WID as a function of age in Fig. 2b.

The frequency of cultural traits, including music preferences, can shift over time due to a process known as 'cultural drift'[69]. This drift has a pronounced effect in smaller populations, causing significant changes in trait frequencies. However, in larger populations, these frequencies tend to remain fairly stable. Our research supports this, as it shows a higher uniformity in music tastes in smaller areas, as indicated by a lower BID (Fig. 1b) and dense concentrations in user vector space (Fig. 1e). As a result, cultural diversity in large population areas may also be preserved for extended periods of time, in turn influencing vertical transmission of cultural taste. Children growing up in large cities will be exposed to a broader spectrum of cultural content and have greater access to a variety of musical training programs, and this exposure can significantly influence their music perception[70]. Our current data, which is a static snapshot, is unable to capture these potential vertical transmission mechanisms. Future research would benefit from examining longitudinal data to understand how individual and regional taste traits evolve over time.

## Limitations

There are several limitations to our study. First, our study is limited to music. Despite music being omnipresent in all human cultures[34], not all music is easily accessible or documented, particularly in cultures with strong oral traditions[71]. This can create a selection bias in favour of music that is registered on globalised media platforms, and subsequently studied. The global music market is also heavily influenced by accelerating globalisation and major music labels that dictate what is broadcast. These influences are often complex and difficult to disentangle. Although similar issues may persist in other cultural studies[67], extending our research to areas such as language, literature, art, societal values, and traditions would help to test the generalisability of the phenomenon.

Second, while our study replicates the trends in different countries best covered on the Deezer platform, the findings are limited to industrialised countries where the majority of the population lives in urban centres (User demographic in Methods), and our causal inference testing is limited to France due to the spatial granularity of the data. With the exception of age and gender, other demographic factors for causal testing were approximated from postcode level averages through inferred users' home locations. Thus, the regional data may be limited in fully capturing the subtle variances among individuals. Moreover, cultural contexts are known to significantly influence music consumption patterns[39], and the user characteristics of the Deezer platform may not perfectly align with the general population's behaviour—although the general demographics appear to be similar (User demographic in Methods). Future studies can incorporate wider global coverage to test if the results replicate in other countries, particularly focusing on those with a larger proportion of rural populations.

Third, our data cover a narrow temporal duration of a 4-week window in March. This particular window was carefully selected because we wanted to minimise potential issues with people travelling during the summer holidays and Christmas periods, when mass population movement can bias our geographical-grounded data[49]. However, it is possible that seasonal patterns at longer or shorter time constants can influence our results, and this can be investigated in future work.

Finally, we acknowledge that there may be other important confounders we did not consider, and that more demographic data could ultimately change the results. Specifically, different cultural norms and disparities among urban and rural areas in their internet connectivity and media exposure may be important factors to further consider. Nevertheless, here we have made an extensive effort to augment a variety of information that is typically overlooked, such as acquiring social connectedness and data on music venues. Our main aim in illustrating the model with a DAG is to transparently lay out our assumptions that can be refuted and corroborated by future researchers to serve as a useful framework.

Determining causal relationships would ideally necessitate intervention experiments[58,59]. However, in the context of our study and many others examining social phenomena, it is often not feasible nor ethical to design experiments manipulating individuals' cultural environments. Thus, past studies have relied on rather simple hypothetical scenarios with small sample sizes to test the causality of group size on cultural diversity and complexity[16–19,22]. Nevertheless, the advent of online experimental platforms and advanced recruitment methods now enables large-scale and cross-cultural studies. Particularly in the domain of music, iterative transmission paradigms have enabled the emergence of musical universals and diversities in controlled lab settings, providing insights into their mechanisms[72–76]. Future experiments could, for instance, allocate participants randomly across different micro-societies[77] to produce or consume a specific cultural medium. These micro-societies can then be manipulated in their size, flow of immigrants (i.e., new participants), and the breadth of each participant's interaction controlled by the network structure—as previously done with research in language[78]. Such future experimental approaches for determining causality can go in tandem with our large observational analyses of real-world behaviour for answering questions about the general mechanisms underlying cultural evolution and transmission.

## Methods
### Music streams
Our research was conducted on Deezer, a globally available music streaming platform present in 187 countries, containing a catalogue of over 90 million tracks. Users' listening behaviour is captured with comprehensive logs, including the date and time of song playback, listening duration, the listener's self-reported age and gender (when provided), preferred language, type of streaming device and network connection, geographical localisation at the city-level derived from a third-party service, and whether the stream is organic (i.e., user explicitly making the choice of music) or algorithmically recommended (Algorithmic streams in Method). We gathered all songs that were streamed in France, Brazil, and Germany between 6 March and 2 April 2023, a four-week period ensuring a balanced representation of weekdays and avoiding holiday seasons. To reduce noise and potential biases in our data, we excluded streams that were played for less than 30 seconds and those over mobile networks, which are unreliable for geolocation tracking. The data was handed to the researchers anonymised and the analysis was not used to derive commercial profiling of any kind. In accordance with the Deezer user agreement, users have consented to their data being used for the purpose of advancing scientific research. The study was approved by the Ethics Council of the Max Planck Society (Application No: 2025_8).

### User sampling
From the sampled streams, we excluded users who streamed less than 100 times within the study window and those who frequently travelled (more than 10 unique geo locations identified). This criterion was set by assessing the general frequency distribution of unique locations per user (Supplementary Fig. 1). The results were also replicated with more loose and strict thresholds (Supplementary Fig. 2). Each user's home location was approximated from the most frequently streamed city-level locations. Using data from GISCO (version year = 2020), French and German users were mapped to Local Administrative Unit (LAU) areas (number of LAU units: France = 34,968, Germany = 11,007). LAUs are building blocks of the NUTS (Nomenclature of territorial units for statistics) and statistical areas, and comprise the municipalities and communes of the European Statistical System (ESS). Brazilian users were mapped to municipality-level areas following the data (version year = 2021) published by the Instituto Brasileiro de Geografia e Estatística (IBGE). This mapping procedure excluded 5.4% of the users who could not be mapped to any of the areas within these geographical boundaries. Finally, we grouped the remaining users in France and Germany by the NUTS3 unit areas (one level higher than LAU) to reduce noise when measuring BID and WID, while keeping the municipality level for Brazil as grouping by the state was too broad. Areas with less than 200 unique users were excluded from our analysis to reduce noise and to ensure anonymity when shared as aggregated-level data. This resulted in 96 and 113 NUTS3 unit areas remaining for France and Germany, respectively, and 218 municipalities of Brazil.

### User demographic
In the final set, 2,544,549 users remained (France = 1,506,899 (47.1% Female), Brazil = 816,101 (33.6% Female), Germany = 221,547 (46.6% Female). A detailed demographic comparison across the countries by their self-reported age and gender and the number of monthly streaming activities can be found in Supplementary Fig. 9. Population census data showed that the majority of the population in these countries live in urban areas (France = 81.5%; Brazil = 87.6%; Germany = 77.6%), which were all above the world average ($M = 62.0\%$, SD = 23.4%), and thus represent characteristics of highly urban countries. To assess how representative our user sample is of the general population, we compared the user demographics to the Eurostat census data on population size, median age, and gender (Socio-demographics in Methods). We tested this for France, given the highest sample granularity and importance in our analysis involving causal testing. When compared across the NUTS3 area population of France, strong correlations were observed between the number of users and census population size ($r = 0.90$, $P < 0.001$), and their median age ($r = 0.61$, $P < 0.001$). However, when stratified by age groups, our user sample consisted of 25% more young (between ages 15–30), 3% more mid-age (31–50), and 30% less elderly (51–80) populations. Our sample

demonstrated a slightly skewed tendency towards having more male demographics (52% male) compared to France's general population (48% male). The gender ratio per area did not significantly correlate ($r = 0.13$, $P = 0.179$), suggesting platform-wise biases in gender.

## Embedding space

**Song embedding.** Many recommendation systems leverage embeddings to efficiently encode the latent relationships between users and content. Embedding is a multidimensional space where objects, such as songs in this context, are represented as vectors in a way that captures the relationships between these objects based on user behaviour, namely, co-occurrences of songs across playlists and listening patterns. In this *space*, closely related songs are positioned near each other. Tracks that share thematic or stylistic similarities tend to also have high co-occurrences. For instance, Adele's 'Someone Like You' and Sam Smith's 'Stay With Me' would be located close to each other within this space. To provide a tractable low dimensional space, Deezer employs the singular value decomposition (SVD)[79] technique based on the co-occurrence of millions of songs. This matrix is then approximated through a factorisation technique to yield a 128-dimensional embedding space[80], capturing the nuanced relationships between songs based on user interactions and thematic links.

**User embedding.** Each user's listening behaviour, represented in the song embedding (Song embedding in Methods), can be summarised to identify a central position in the space, defined as the average of all songs they have listened to in the past 28 days. This results in a 128-dimensional user vector, which can then be used to construct a user embedding space—similar to how relations between songs were converted into song embeddings. In the user embedding space, users with similar music preferences (often corresponding to fans of a particular genre) are positioned nearer to each other, while those with distinct tastes are placed far apart.

## Measuring diversity

**Between-individual diversity (BID).** To quantify the diversity found in a given population, existing research has commonly applied measures like the Gini coefficient[81], Simpson's index[82], or Shannon's entropy[83]. However, these measures have been criticised for their arbitrary scales, making comparisons between results challenging. As a solution to this, Hill's number (also known as the effective number of species) has become an increasingly popular method to quantify the diversity of a species assemblage. It allows for standardised comparisons by encompassing various diversity metrics through varying the order of a single parameter $q$[46,84]. Originating from ecology, Hill's number treats the abundance of species in an ecosystem (or field site). In our use case, we treat each song as a *species* and the abundance of songs streamed in a geographical area as their ecosystem (i.e., NUTS3 areas of France and Germany, municipalities of Brazil). It essentially quantifies how diverse the music consumption is in a given area, indicating whether individuals listen to a wider range of songs or concentrate on a few popular ones. From the 1000 music streams we sample in each bootstrap (Statistical analysis in Methods), a Hill's number of 900, for instance, implies that the diversity is equivalent to having 900 songs that are equally represented in the dataset—that is, how many equally common songs would produce the observed diversity.

We calculate between-individual diversity (BID) of music engagement as ($^qD$), expressed as:

$$^qD = \left( \sum_{i=1}^{s} p_i^q \right)^{\frac{1}{1-q}} \tag{1}$$

Here, $q$ defines the order of Hill's number, where higher values of $q$ emphasize the contribution of rare songs, while lower values of $q$ focus

on the abundance of popular songs. The $S$ represents the total number of unique songs, and $p$ signifies the relative abundance of each song.

The Hill's number of order $q = 1$ is then defined as the limit of the expression in Eq. (1) as $q$ approaches 1:

$$^1D = \frac{1}{\prod_{i=1}^{R} p_i^{p_i}} = \exp\left( - \sum_{i=1}^{R} p_i \ln(p_i) \right) \tag{2}$$

which essentially becomes the exponential of the Shannon entropy in natural logarithms. In our analysis, we set the order of $q$ to be 1 a priori, but results were also robust to other values of order $q$ (Supplementary Table 3).

Given that algorithmically recommended content can bias the calculation of BID, we excluded all algorithmic streams and used only the organic streams for the main analysis of Fig. 2 (Algorithmic streams in Methods). However, additional analyses that do not exclude algorithmic streams, as well as analyses considering them separately, can be found in Supplementary Table 2. For causal testing (Fig. 3), we did not exclude the algorithmic streams as we include this as a potential confounder (Algorithmic streams in Methods).

**Within-individual diversity (WID).** To assess an individual's diversity of musical engagement, we employed the Generalist-Specialist Score (GS-Score), a previously validated metric in user music exploration and discovery studies[45,49]. The GS-Score computation relies on the high-quality song embeddings of Deezer that summarise relationships between songs as high-dimensional vector representations (Song embedding in Methods). First, a user's ($\mu$) centroid position in the song embedding is defined by computing the mean vector ($\vec{\mu}$) of all the songs the user has listened to within the last 28 days. Next, the cosine similarity is calculated between $\vec{\mu}$ and a randomly selected song ($\vec{s}$) the user listened to, weighted by the number of times they have listened ($w_s$). This approach ensures the measure is not sensitive to the number of songs that the user has listened to. Moreover, WID is computed based solely on the user's explicitly chosen content, and thus does not include algorithmically recommended streams. The resulting GS-Score ultimately captures the user's radius of coverage in the song embedding, formally written as:

$$GS(\mu) = \frac{1}{\sum_s w_s} \sum_s \frac{w_s \cdot \vec{s} \cdot \vec{\mu}}{|| \vec{s} || \cdot || \vec{\mu} ||} \tag{3}$$

If a user is a 'specialist', they would have a smaller radius, indicating a more focused interest, whereas a 'generalist' would exhibit a wider radius, indicating a broader range of music engagement. As such, unlike BID, WID does not rely on categorical grouping but instead captures the distances within the vector space. To make this score consistent with the direction of our BID measure, we inverted the score ($1 - GS(\mu)$) and normalised the value to range between 0 and 100, where 100 represents maximal WID.

**Dispersion in user embedding.** The frequency-based measure of BID (Between-individual diversity (BID) in Methods) potentially overlooks the relationships between the songs. An alternative approach that can provide a higher-level characterisation is to measure how misaligned users are in their music preferences by utilising distances within the user vector space (User embedding in Methods). As described previously, users with similar musical preferences, or taste, will have a shorter distances within the space, while users with distinct tastes will be far apart. At the geographical area level, we can compute the pairwise distances across all users and measure the dispersion, or the radius, as an indicator of the diversity of music preferences of the given area. Formally, this user dispersion is computed by taking the population variance in the pairwise distances of users from a given

area, written as:

$$\text{population variance} = \frac{1}{N} \sum_{i=1}^{N} (x_i - \bar{x})^2 \qquad (4)$$

Where $x_i$ is the cosine similarities of bootstrapped pairs of user vectors, and $\bar{x}$ is the mean cosine similarity across all users. This approach of leveraging vector space and measuring the dispersion is analogous to the method used for measuring WID using the GS-Score computation (Within-individual diversity (WID) in Methods).

## Socio-demographics

Our DAG model includes seven socio-demographic confounders. We outline the data sources for each and discuss the rationale behind their inclusion, drawing on existing literature.

**Age and gender.** Studies have shown how one's musical exploration[45] and preferences[51] are influenced by age, demonstrating that one's music taste generally consolidates during adolescence. Research has also shown significant differences between the consumption patterns of males and females, noting that male users on online platforms tend to consume more diverse and niche content[44,85]. We thus include the user's self-reported age and gender as potential confounders influencing music consumption diversity. Among the users who provided information (89.7%), age was computed based on the self-reported birth date. When registering, a user could specify their gender from the following options: 'Male', 'Female', 'Unknown', 'Non-Binary', 'Other', or left blank. For simplicity in our DAG analyses, we only included users with ages above 18 and below 65, and who self-identified as Male or Female (13% excluded; see Supplementary Fig. 9 for demographic distributions). This was to reduce noise as there were only a small number of individuals per area outside of this criterion.

**Immigration, education, and income.** Immigration or contact between populations has been shown to act as a vibrant channel for cross-cultural exchange, introducing fresh perspectives to a culture, and fostering innovation and complexity[10,68]. In parallel, one's educational attainment and economic status have been shown to be tightly linked with their cultural inclination. Rooted in the cultural omnivore theory widely debated in the social sciences[32], numerous studies have observed that the societal elites seek a broader spectrum of cultural experiences[53–55]. Considering this past literature, we include dimensions that can indicate social class and immigrant status. The data were drawn from Cagé and Piketty (2023)[86], who aggregated and made openly available electoral and socio-economic data from the municipalities in France, with longitudinal records dating back as early as the 18th century. These data are collected from the electoral reports digitised in national archives such as the L'Institut National de la Statistique et des études économiques (INSEE)—the National Institute for Statistics and Economics Studies of France. Among various socio-demographic indicators they collect, we focused on immigration, education and income at the level of communes that divide France into over 35,000 area units, which corresponds to the granularity of postcode resolution. We used these area averages as proxies for each user's attributes (see Supplementary Fig. 11 for geographical map visualisations). Specifically, we used the most recent data from 2022 on: (1) percentage of immigrants (pimmigre2022) from the 'naticommunes' dataset, (2) percentage of residents with bachelor's degrees (pbac2022) from the 'diplomescommunes' dataset, (3) and average per capita income (revmoy2022) from the 'revcommunes' dataset (for an accurate description of the columns and source of the data, see their appendix material[86]). Assessing by the Q-Q plot, we observed that income and immigration percentages across the municipalities were highly skewed in their distribution, thus log transformation was applied (Supplementary Table 5).

**Musical venues.** Cultural activities concentrate in places like cities and towns[87]. Residents of metropolitan and large cities have easier access to diverse cultural offerings compared to their rural counterparts, which may subsequently influence their cultural engagement. To approximate the amount of access available to cultural events, we gathered information about the musical venues at the NUTS3 level using SongKick's database, a popular global concert discovery service. Using their API, we initially queried 10,000 venues in France in August 2023. After excluding venues without geolocation information, 6618 venues remained (see Supplementary Fig. 12 for geographical map visualisation).

**Algorithmic streams.** Recent studies on the effect of algorithmic recommendations have shown a direct link to users' diversity in music consumption, suggesting that individuals who tend to have more diverse tastes rely less on algorithmic recommendations and engage in more organic exploration[49]. To test for possible differences in algorithmic recommendations usage across population areas, we sub-sampled 10,000 individuals at random per area in our dataset and across the three countries. We then computed the proportion of their algorithm-driven music streams over all streams, sub-grouped by age and gender. Algorithmic streams include listens such as auto-played next song after listening to a song or an album, and personalised recommendations. Organic streams, on the other hand, are users' explicitly chosen content such as searches and songs already included in their music library. In France and Brazil, there was a moderate to strong negative relationship between algorithmic stream proportion and population size across all age groups (see Supplementary Fig. 3 for statistics). This suggests that individuals living in large metropolitan areas tend to rely less on the use of algorithmic recommendations; however, the effect was small. There was also an effect of gender across all three countries, where male users tend to use algorithms more than female users. Given such apparent differences in algorithm usage by area size, we include the proportion of algorithmic streams as a confounder in causal testing. Although this algorithmic bias might affect our BID measure (Supplementary Table 2), it does not directly impact the WID since it is anyhow calculated solely based on users' explicitly chosen content (i.e. organic streams). Nonetheless, an indirect pathway that could potentially influence the WID may exist, and we represent this in the DAGs (Fig. 3).

**Social connections.** Engaging in international connections can significantly influence one's exposure to diverse content by providing access to cultural content beyond their own cultural sphere[88,89]. Increased international connections may also suggest extensive travel experience or a background of living abroad. Recent research has also found that individuals tend to broaden their preferences and interests towards the cultural influences of the places they visit[90]. We used the publicly available dataset released by Meta (reference period: 13th of October, 2021) to approximate the number of international Facebook friends one has at the level of NUTS3 units. The Social Connectedness Index (SCI), first introduced by Bailey et al.[91] uses an anonymised snapshot of all active Facebook users and their friendship networks to measure the intensity of social connectedness between locations. Users are assigned to locations based on their information and activity on Facebook, including the stated city on their Facebook profile, and device and connection information. Formally, the *SCI* between two locations $i$ and $j$ is defined as:

$$\text{SCI}_{i,j} = \frac{\text{Connections}_{i,j}}{\mu_i \times \mu_j} \qquad (5)$$

Here, $\mu_i$ and $\mu_j$ represent the number of Facebook users in locations $i$ and $j$, and $\text{Connections}_{i,j}$ is the total number of Facebook friendship connections between individuals in the two locations. This metric

effectively captures the relative probability of a Facebook friendship link between locations. To quantify the amount of international social connections, we summed the SCI paired with all other areas around the world that are not from the same country (i.e., France). We then normalised the scale and applied a log transformation to account for the skewness (Supplementary Table 5). To validate how Facebook user demography compares with the Deezer users in France, we collected data from a private company (https://napoleoncat.com) that gathers country-level Facebook user demographics, for the same month as our sampling window in March 2023. Demographics were similar for both females and males across the age groups (same Spearman correlations for both genders = 0.86 [0.29, 0.98], $P = 0.08$), with adolescents taking up the largest proportion in both platforms (see Supplementary Table 5 for raw values).

### Causal inference

The DAG was illustrated and evaluated using the 'dagitty' R package[92]. A step-by-step procedure for causal testing that we applied is described in full detail in Supplementary Note 1 but summarised here. We first checked the consistency of our data with the DAG models and the robustness of various versions of the candidate models[60,93]. Our ultimate model successfully passed several implied independence tests, which evaluate whether certain variables in the model are truly independent of others when considering the values of a different set of variables. This testing is crucial as it validates the model's representation of relationships and lends support to our DAG hypotheses. All variables were normalised with Z-scores for effect size comparisons. Variables that did not follow a normal distribution were log-transformed (Supplementary Table 5).

To control for confounders, we used propensity scores to adjust for group differences in users living in different size areas using the 'WeightIt' R package[94]. The propensity score condenses all observed covariates into a single metric[95]. Acting as a balancing measure, it aims to equalise the distribution of confounders between individuals across the groups. Each individual is assigned weights using inverse probability weighting (IPW)[96], which determines how much they 'contribute' to the group. Consequently, it enables the simulation of a quasi-randomised scenario to facilitate causal inference testing[58,93]. To obtain estimates of the causal effect, a weighted generalised linear model (GLM) was fitted to model the outcome of interest. To quantify the uncertainty associated with this estimate, we conducted bootstrap simulations on the entire sampling and weighting procedure (Statistical analysis in Methods).

### Statistical analysis

All hypothesis tests were conducted using a bootstrap with 1000 replications to derive the mean, with the exception of when computing BID and WID. For BID, 1000 unique streams were drawn from each area, while for WID, 100 unique individuals were drawn for each bootstrap. We sample a fixed amount of streams (BID) and a fixed amount of users (WID), so that larger areas are not a-priori overrepresented. Confidence estimates were derived from the 2.5% and 97.5% quantiles of the bootstrap means. Pearson and Spearman correlation coefficients were adjusted for multiple comparisons using the Holm method[97]. One-way ANOVA was used for comparison across groups and post-hoc Tukey's test $p$-values were also adjusted for multiple comparisons. Cohen's $d$ was used for effect size estimates[98]. Analysis was conducted using R (version = 4.3.3). Unless explicitly mentioned, all stats were computed using the 'stats' package in base R and custom scripts (Code availability).

### Reporting summary

Further information on research design is available in the Nature Portfolio Reporting Summary linked to this article.

## Data availability

The raw individual listening data used in this study are protected and are not available due to data privacy laws and our agreements with streaming services. The aggregated and anonymised data generated in this study have been deposited to Zenodo (DOI: 10.5281/zenodo.15255221)[99] and openly available under the same DOI and on GitHub (https://github.com/harin-git/mus-div). Population size data were obtained from Eurostat census data available at https://ec.europa.eu/eurostat/web/main/data/database. Demographic data on age, gender, immigration, education, and income were compiled from longitudinal census data by Cagé and Piketty (2023), available at https://unehistoireduconflitpolitique.fr/telecharger.html, with raw data originating from INSEE. Social connectedness measurements were based on the Social Connectedness Index (region pair Facebook friendship connections) released by Meta, available at https://data.humdata.org/dataset/social-connectedness-index. Music venue information was collected using the Songkick API (https://www.songkick.com/developer), which requires developer access approval.

## Code availability

All analysis scripts, including the working scripts and plots used for the study, are available at https://github.com/harin-git/mus-div.

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

## Acknowledgements

The authors would like to thank Minsu Park, Ofer Tchernichovski, Mason Youngblood, Bruno Sguerra, and Elena V. Epure for their feedback on the initial version of the manuscript. HL was funded by the International Max Planck Research School on Neuroscience of Communication.

## Author contributions

H.L., R.H. and M.M. jointly conceptualized and designed the study. H.L. performed the data analysis and drafted the original manuscript. N.J. provided analytical support and contributed to manuscript editing. R.H. and M.M. jointly supervised the project. All authors participated in manuscript revision and approved the final version.

## Funding

## Competing interests
The authors declare no competing interests.
