## [Transparent Peer Review file · Nature Communications]

Mechanisms of Cultural Diversity in Urban Populations

Corresponding Author: Mr Harin Lee

Version 0:

Reviewer comments:

Reviewer #1

(Remarks to the Author)

The study explores the factors contributing to cultural diversity within large urban populations by examining the music-listening habits of over 2.5 million individuals in France, Brazil, and Germany. The study tests two hypotheses: the "demographic mixing" hypothesis, which suggests that diversity stems from demographic variances among the population, and the "cultural breadth" hypothesis, which proposes that individual exposure to diverse cultural inputs broadens personal cultural tastes. The article presents a comprehensive analysis showing that while demographic factors such as age contribute to cultural diversity, they do not fully explain the patterns observed. It concludes that the greater opportunities for cultural interaction and exchange in urban settings play a significant role in enhancing cultural diversity.

The manuscript is well-written, and the methodology is sound and clearly described. However, I find the connection between the study's findings and the broader implications for the field of cultural evolution somewhat tenuous. The main aim, as stated in the Abstract and first paragraph of the Discussion, is to explore the mechanisms that shape cultural diversity in highly populated areas ("Does this diversity stem from [...] [...] we sought to decipher the mechanisms and factors contributing to cultural diversity"). Yet, what the study predominantly reveals are statistical associations and, to some extent, causal links between demographic variations and diversity in listening behaviors. This form of diversity in cultural consumption—while interesting and well-analyzed—does not clearly or directly relate to the diversity of cultural forms and elements, whose connection with population size was prioritized in previous cultural evolution studies. In my opinion, the term 'cultural diversity' seems to be used ambiguously, somewhat conflating consumption patterns with broader cultural forms without clearly elucidating their interaction. The attempt to bridge this gap in the Discussion section remains underdeveloped and would benefit from a deeper integration with cultural evolutionary theory. Questions remain as to which evolutionary mechanisms—be it biased selection, drift, mutations, guided variation, or blending inheritance—mediate the observed relationships. These mechanisms' roles in influencing cultural traits' frequency and emergence are crucial for aligning the study more closely with significant cultural evolutionary frameworks and enhancing its impact in the field.

More specific comments and questions follow below.

Introduction.

The authors define "cultural diversity" in terms of variation in listening habits or music engagement. This differs from the traditional understanding, which encompasses variation in human cultural traits such as the structure and forms in music or language (Lomax, 1959; Rzeszutek et al., 2012). A more comprehensive explanation of how these two forms of diversity are related would clarify the study's contribution to cultural evolution.

The cultural breadth hypothesis posits that continuous exposure to diverse cultural inputs broadens one's cultural taste. Is this hypothesis supported by experimental evidence from psychological studies, or is it merely theoretical? A brief explanation with references to experimental findings would strengthen this section.

"We collected data from France, Brazil, and Germany, which are the three countries with the largest percentage of Deezer users relative to their general populations." How do these populations represent global urban versus rural dynamics? A discussion on their representativeness in the context of global urban-rural dynamics would be valuable.

While music is an ideal test-bed for examining cultural diversity due to its universal prevalence and cultural distinctiveness, potential drawbacks exist. What are the limitations of focusing solely on music? How might findings generalize to other

cultural domains? Addressing these questions would provide a more balanced view of the study's scope.

Results

I did not completely understand how BID and WID were calculated. Based on the information provided in the manuscript, it appears that 10,000 unique music streams were sampled for each bootstrap simulation from a subset of 100 random users. Is that correct? Or were the 100 users to be bootstrapped (Fig. 2b)? I am confused by the use of "respectively" in the Statistical Analysis section. This point should be made clear.

In Fig 1c, do the significant correlations hold when music streams are grouped by artist and genre in control analyses? This information is not found in the supplementary material but should be included to ensure transparency and robustness of the findings.

Supplementary Fig. 4 shows significant correlations between BID and population size for algorithmically recommended streams. Do these correlations persist when using alternative groupings, such as by users and genre? Including this analysis in the Supplementary Material would provide a more comprehensive understanding.

The analysis is based on a four-week window of listening behavior. Discuss the temporal stability of these patterns and whether short-term trends or seasonal variations in music popularity could influence the observed BID levels. Acknowledging the limitations of this timeframe in capturing the full spectrum of temporal variability on BID levels, particularly in urban areas, is essential.

How do Deezer's recommendation algorithms, which influence music exposure, affect the observed diversity metrics? Discuss the potential for algorithmic biases or personalized recommendations to skew BID, especially among heavy users who depend on these recommendations for music discovery.

In Figure 1D, only two regions (Paris and Orne) were used to measure regional user dispersion. Is there a reason for not including more samples to support this approach's validity? Including additional examples of both highly populated and smaller areas in the Supplementary Material would strengthen the analysis.

WID

Regarding the WID analysis by age, how much of the observed trend can be attributed to biases in the population structure? For instance, if the sample predominantly consists of participants in their 20s, with fewer older users (as indicated by larger error bars with increasing age), this potential bias should be discussed.

DAG

While the study controls for several socio-demographic variables, other potential confounders, such as media consumption habits or cultural norms, may influence the relationship between population size and diversity. Differences in internet connectivity and media exposure between urban and rural areas could also be factors. If obtaining this information from statistical institutes is not feasible, these limitations should be acknowledged.

The study uses inverse propensity weights (IPW) to balance confounding variables across groups. How effective is this weighting method, and could there be potential issues with extreme weights or unbalanced covariates affecting the results?

The study focuses on France due to its high spatial granularity and demographic data. Discuss more thoroughly how generalizable the findings are to other countries or cultural contexts, particularly those with different population structures or music consumption patterns.

(Remarks on code availability)

Reviewer #2

(Remarks to the Author)

The authors provide a compelling test of two competing hypotheses about the super-linear increase in cultural diversity in (large) cities using large-scale behavioral data (music streaming). The paper is clear, well written, and all in all compelling - the methods are adequate and there are multiple complementary analyses that help squeeze relevant information out of their datasets.

My only points of contest are two:

1. The authors cannot show that age is the relevant variable explaining differences in BID/WID. In order to do that, they would need to have measurements over time. As of now, their results could be just a result of cohort (people born in the 50s having overall less diverse musical taste than people born in the 80s even conditioning on age). Please make this clear.
2. "...there was no discernible difference in the cultural breadth of individuals living in medium (inhabitant size around

500,000) versus large regions (more than 1.2 million), suggesting an upper limit in the extent that size fosters cultural exploration. This limit is likely reached when a region has sufficient infrastructure to support diverse cultural experiences." I think it's a bit premature to over-generalize any conclusions on the basis of three cities alone - it won't harm some nuance in these statements.

(Remarks on code availability)

Reviewer #3

(Remarks to the Author)

The manuscript NCOMMS-24-24231-T investigates a question of broad interest regarding the drivers of cultural diversity – do larger populations support cultural diversity because they contain individuals from more varied backgrounds, or individuals with more varied tastes? The authors test their predictions using a very large, high-resolution dataset based on music streaming activity in over 2.5 million listeners from three countries. Their major findings are as follows: 1) that both within- and between-individual measures of diversity in listening habits increase with population size and 2) demographic differences between larger and smaller populations partly explain these trends. The role of population size and demographics in cultural diversity is a major area of interest in cultural evolution, a highly interdisciplinary field intersecting with anthropology, archaeology, psychology, biology, linguistics and more. This paper therefore makes a potentially important contribution to a very broad field of study.

In general, the methods of this study appear sound and well justified, with careful consideration of potential confounds. We have, however, a number of queries and suggestions for the authors to consider concerning their methodology. We cannot necessarily comment on all of the aspects of the statistical methodology and accept that some of our questions may reflect our lack of familiarity with some of the specific techniques included in the paper. We are experienced in methods commonly used in cultural evolution and in some music information retrieval techniques, but some aspects of the methodology (particularly directed acyclic graphs) are less familiar to us. However, publications in such a broad interest journal should be as accessible as possible to researchers from a wide range of fields and so additional clarification on key aspects of the methods would be beneficial in any case. We appreciate that there is limited space in the main text to address all our comments in full, however, in some cases all that is required is rewording and/or clearer signposting to relevant areas of the Supplementary Information.

The two main issues we have identified are as follows:

1. Diversity measures

The cultural diversity measures could be more clearly explained, particularly for the benefit of broad audiences. The between-individual diversity (BID) measure has been adapted from species diversity measures used in ecology, and a little more explanation of how it translates to cultural diversity would be helpful. For example, if songs are analogous to species (L476), what is the cultural equivalent of an ecosystem – a geographic region? Similarly, the within-individual diversity (WID) measure could be explained in a more user-friendly fashion – phrases like 'general song embedding space' (L509) are not likely to be intelligible to broad audiences.

As the authors note (L532-533), one important limitation of the BID measure is that it captures population-level diversity in the songs that are listened to, but not other dimensions of diversity (e.g. genre/stylistic diversity) which are potentially more meaningful. The 'dispersion in user embedding' measure addresses this issue but as before it could be more clearly explained – the relevant section of the Methods (L514-L543) is technically dense and challenging to understand (even for those with some experience in music information retrieval). It is also not completely clear to us whether this issue also affects the within-individual diversity measure. Readers can find some comparisons of song-, artist- and genre-level measures of BID in the SI but these are only very briefly referred to in the main text (L134). It appears that effects of population size are weaker for genre- than song-based BID measures, but only the strongest correlations are reported in the main text (L133) – the authors should be more transparent about this. Finally, it's not clear to us why the results are repeated using the dispersion in user embedding measure only in France, with results displayed only for two selected French regions in Figure 1. Instead, we suggest that correlations with all diversity measures are reported for all three regions in SI Table 1 so that readers get a full picture of the results and can easily compare strength of effects for different diversity measures.

2. Algorithmic recommendations

The premise of the study, as established in the introduction, is that cultural diversity is the result of diversity between individuals in their experiences and/or within individuals in their preferences. However, hypotheses are tested using measures of cultural diversity derived from user behaviour on a music streaming platform, which is to a significant extent influenced not only by users' own predilections but by the choices made available to them by algorithmic recommendations. The authors do mention this issue but only very briefly (L425, 149). I think given the importance of recommender algorithms in user behaviour, it would be worth clarifying this issue further. It's particularly important to consider whether algorithmic recommendations may bias the effects – for example, do people streaming music in larger metropolitan areas receive more diverse algorithmic recommendations? In the Supplementary Information, the authors present a comparison in the relationship of a single measure of BID to population size between 'organic' and 'algorithmic' streams which shows qualitatively comparable results although effects are a little stronger when algorithmic streams are included, which suggests that they may introduce some biases in favour of the hypothesis. We suggest that the authors either 1) remove algorithmic

streams from all of their analyses or 2) include a side-by-side comparison of all key results between organic and algorithmic samples in the Supplementary Material.

We have a further list of shorter, but still important, queries and suggestions as follows:

- The observations in the analyses are not necessarily independent due to spatial effects: neighbouring regions are probably more similar in cultural diversity/population size than distant regions. Do the authors need to consider and control for spatial autocorrelation?
- In the introduction, the two hypotheses are presented as separate, but it seems that the second depends on the first – even if people living in urban areas are more open to diverse cultural activities, without a ‘source’ of cultural diversity (i.e. individuals from diverse backgrounds, sharing diverse cultural ‘products’), then diversity surely cannot increase with population size.
- In the data availability statement it appears that variables are maintained in separate datasets (i.e. github repository, census data, Songkick database, etc.). This makes it more challenging for readers to replicate the analyses presented in the paper – we would suggest that variables are combined into single datasets as much as possible.
- We assume that Deezer streamers have consented to their data being included in research studies like this one through user agreements, but it would be good to be explicit about this.
- The section on casual inference (L618-635) is tough going for general readers – many key concepts (e.g. ‘implied independence’) are likely to be unfamiliar. It’s also not clear to us how the covariates were ‘condensed’ into a single ‘propensity score’. It may, further, be helpful to explain some of the limitations of causal inference for observational data. Could further detail and explanation be provided in an extended methods section in the SI?
- Results are interpreted in terms of urban versus rural differences (e.g. L134-136), while strictly they are effects of population size (which obviously differs between urban and rural areas, but population size is only a proxy for urban/rural differences).
- The authors describe ‘cross-cultural differences’ in diversity (L139) but they are referring to differences between three large, multicultural nations (France, Germany and Brazil) so perhaps ‘between-country differences’ would be more appropriate. It is also not clear whether these comparisons account for socio-demographic differences between the countries.

Finally, we have identified a list of minor suggestions/typographic errors:

- L56: should ‘megapolis’ be ‘megapolises’?
- L68-69: Street et al. (2022) found that tunes played by larger communities have more varied versions, but these tunes tend towards intermediate (not higher) complexity.
- L127: ‘resulting as’ should be ‘resulting in’
- L146: the term ‘control experiments’ may be confusing for readers as all analyses are based on observational data. Perhaps ‘additional analyses’ or similar would suffice here.
- L158-159: the phrase ‘dispersion amount of regional users in each region’ is challenging to understand
- L376: typo (‘musicl’), also the Street et al. study looked at melodic rather than stylistic diversity
- L450: ‘this resulted as’ should be ‘this resulted in’
- L478: ‘specie’ should be ‘species’
- L483: the ‘q’ parameter in Hill’s number could be more clearly explained – is it assumed a priori or estimated from the data?
- L551-552: it would be interesting to know what the expected differences are in streaming behaviour between genders
- L557: ‘self-identified themselves’ is redundant, you can just say ‘self-identified as’
- L558: the exclusion of participants over 65 and those identifying as other than male/female could be better justified
- L594-607: measures of social connectedness are based on Facebook data, but we wonder if this matches well with the streaming user demographics, especially given that Facebook use is declining among younger demographics
- L643: what method of adjusting for multiple comparisons was used?

(Remarks on code availability)

Version 1:

Reviewer comments:

Reviewer #1

(Remarks to the Author)

I commend the authors for their thorough work on this revision. The focus of the research and the methodology employed are now much clearer, thanks also to the valuable comments from the other two reviewers. This study makes a significant contribution to our understanding of how demographic factors may influence diversity in music consumption. I have made a few minor comments on points in the Discussion section, before my final consideration.

- first paragraph in the Discussion section:

Clarify again what aspect of 'cultural diversity' the study addresses (i.e., music consumption). This may be useful for readers who only skim through the key structural points of the article, such as the abstract and the first paragraph of the discussion, to assess whether the paper aligns with their interests.

"[...] We observed an increasing trend of between-level diversity [...]"

Between-individual diversity?

"[...] This provides direct empirical evidence [...]"

The term 'direct evidence' should be used cautiously in this context. Causal inference tools, while valuable, have limitations in providing conclusive evidence without experimental manipulation --I see this point was already made by another reviewer, but it is critical to stress. The study would benefit from a more thorough discussion of experimental designs that could address these limitations, such as transmission chains or more complex microsocieties. These methods can indeed provide critical information on the mechanisms behind the origins and persistence of cultural variation (Mesoudi 2007), in all its forms (production, perception, consumption).

While the Limitations section briefly mentions these approaches, it lacks a precise description of how they could overcome the current methodological constraints. In this regard, a short section primarily focuses on mass online designs (e.g., Anglada-Tort et al. 2023), which, while offering large sample sizes and cross-cultural implementation, may lack the rigorous experimental control of laboratory designs. To strengthen this aspect, the authors should consider *only briefly* expanding on how laboratory methods could be applied specifically to their research questions, and how that could address current limits (e.g., manipulating critical variables, randomly assigning micro-societies with different demographical characteristics to different experimental and control conditions, etc.). Additionally, it would be beneficial to cite examples of such methods already implemented in the musical domain, such as Ravnani et al. 2016, Lumaca et al. 2017, Verhoef and Ravnani 2021, and Popescu and Rohrmeier 2022. These four examples provide concrete illustrations of how mechanistic questions on the origins of musical diversity (and universals) have been addressed using more controlled experimental designs.

refs

Mesoudi, A. (2007). Using the methods of experimental social psychology to study cultural evolution. *Journal of Social, Evolutionary, and Cultural Psychology*, 1(2), 35.

Ravnani, A., Delgado, T., & Kirby, S. (2016). Musical evolution in the lab exhibits rhythmic universals. *Nature Human Behaviour*, 1(1), 0007.

Lumaca, M., & Baggio, G. (2017). Cultural transmission and evolution of melodic structures in multi-generational signaling games. *Artificial Life*, 23(3), 406-423.

Verhoef, T., & Ravnani, A. (2021). Melodic universals emerge or are sustained through cultural evolution. *Frontiers in Psychology*, 12, 668300.

Popescu, T., Walther, J., & Rohrmeier, M. (2022). Building blocks of tonality emerge from transmission chains with random melodies.

"Using causal inference tools, we found that demographic factors largely contribute to diversity in the most populated areas. However, they did not fully account for the observed trend of increasing diversity with population size. This potentially suggests that cultural diversity found in large metropolises is not only due to demographic mixing, but also results from more frequent cultural interactions and exchanges that these larger cities enable."

The discussion on the relationship between population size and cultural diversity could be more nuanced and comprehensive. While the authors touch on an important aspect - that larger populations facilitate more frequent interactions - this explanation feels somewhat rushed. The authors should consider expanding briefly this section to include additional mechanisms through which population size might influence cultural diversity. One example is global connectivity: Larger cities often serve as hubs in global networks, facilitating the introduction and integration of international cultural products into local traditions. This enhanced connectivity can significantly contribute to cultural diversity.

"Cultural traits resulting from production and consumption interact intimately through feedback cycles, which are common in complex systems, including culture and economy (43,70,71)."

Consumption patterns may not only influence the range of what is produced but can also inspire the blending of genres or the emergence of entirely new forms (*innovation*). When consumers show interest in more diverse cultural products, producers may respond by experimenting with new combinations of styles and ideas to cater to that demand. This increased diversity in production patterns could then further enhance diversity in consumption and/or perception, reinforcing the feedback loop discussed in the manuscript.

(Remarks on code availability)

Reviewer #3

(Remarks to the Author)

I am really pleased to see that the authors have engaged in depth with all Reviewers' comments and made extensive revisions to their manuscript, which overall strengthen the results and conclusions. Overall, this paper tests fundamental predictions about the drivers of cultural diversity and obtains results which are of broad interest to those in diverse research fields across the social and life sciences using a complex, carefully conducted set of analyses and I would be pleased to see this article published in Nature Communications.

I now have just a few, minor changes to suggest:

- Title: The new, shortened title does not quite make grammatical sense to me – I think you need to add the word 'understanding', 'investigated' or similar to the beginning.
- Abstract: Should '250 million real-world listening behaviours' read '250 million real-world listening events'?
- There are some minor language and grammatical issues remaining throughout, e.g. 'faster pace lifestyle' should be 'faster paced lifestyle', 'show to expand with population size', 'ominovorousness' should be 'omnivory'
- Is it possible to explain the units of BID in an intuitive way? The per-country means (last paragraph of p4) are challenging to interpret.
- The way that 'song embedding' is explained is now less technical, but perhaps too abstract. It may be helpful to clarify that the 'mathematical space' is multidimensional and to explain what particular aspects of song style these dimensions capture (e.g. which rhythmic, harmonic or melodic etc. variables are included, end of p.17/start of p.18). Confusingly, in the next sentence 'Deezer employs the singular value decomposition...' it sounds like the measure of song similarity is actually based not directly on stylistic similarities but rather the co-occurrence of songs in user-generated playlists.
- When describing Hill's number, it would be helpful to clarify that 'assemblage' = 'species assemblage' (p. 18)
- Please mention and cite and specific R packages that were used in the analyses (p. 24)

(Remarks on code availability)

Version 2:

Reviewer comments:

Reviewer #1

(Remarks to the Author)

The authors have addressed all my points and suggestions. I have no further comments.

Massimo Lumaca

(Remarks on code availability)

Reviewer #3

(Remarks to the Author)

I am pleased to see that all my remaining comments have been satisfactorily addressed. I very much support the publication of this article in Nature Communications. I only have a handful of grammatical/wording changes to suggest at this stage:

- Line 4 of the abstract might be clearer if phrased: 'could not discern how individual factors contribute to collective cultural outcomes'
- Line 7 of the first paragraph of the introduction: remove 'show to'
- Line 1 of page 3: I think this should read 'with underlying selection pressures', and you can remove 'the' before 'cultural evolutionary processes'
- Line 4, second paragraph of p10: should read 'and the greatest availability of...?'
- Fig 3 legend: here a little further clarifications would be appreciated by non-specialist readers, including what is meant by an 'exposure' variable and what the thick black arrows in panels a & b indicate
- Line 6, second paragraph of p. 14: perhaps remove the word 'naturally' as this might be misinterpreted as meaning 'innate' or similar.

(Remarks on code availability)

Response to the Reviewers

Aug 16, 2024

Dear Reviewers,

We greatly appreciate the careful and constructive suggestions for our paper entitled “*Tracing the mechanisms of cultural diversity through 2.5 million individuals’ music listening patterns*”. We have fully addressed all the issues raised by the reviewers and revised the manuscript as provided in a detailed response below.

To facilitate an easier task for the reviewers, we labelled each point raised by the reviewers and their comments as **bold** text. Our response is placed just under as normal text and reference to changes in the revised manuscript as *italic* text including line numbers in brackets. In the revised manuscript, all changes are **highlighted in yellow**. In addition, we provide a clean version with all changes resolved.

In general, we edited the manuscript to improve on clarity when possible. We revised the paper title by shortening to “Mechanisms of cultural diversity through 2.5 million individuals’ music listening patterns”. We also refrain from using the term “region” when describing geographical units as this can cause confusion, instead consistently refer to as “area”.

As a result of addressing these issues, we believe the manuscript significantly improved.

Sincerely,

Harin Lee, Nori Jacoby, Romain Hennequin, Manuel Moussallam

Reviewer 1

R1.1: The manuscript is well-written, and the methodology is sound and clearly described. However, I find the connection between the study's findings and the broader implications for the field of cultural evolution somewhat tenuous. The main aim, as stated in the Abstract and first paragraph of the Discussion, is to explore the mechanisms that shape cultural diversity in highly populated areas (“Does this diversity stem from [...]” “[...] we sought to decipher the mechanisms and factors contributing to cultural diversity”). Yet, what the study predominantly reveals are statistical associations and, to some extent, causal links between demographic variations and diversity in listening behaviors. This form of diversity in cultural consumption—while interesting and well-analyzed—does not clearly or directly relate to the diversity of cultural forms and elements, whose connection with population size was prioritized in previous cultural evolution studies. In my opinion, the term 'cultural diversity' seems to be used ambiguously, somewhat conflating consumption patterns with broader cultural forms without clearly elucidating their interaction. The attempt to bridge this gap in the Discussion section remains underdeveloped and would benefit from a deeper integration with cultural evolutionary theory. Questions remain as to which evolutionary mechanisms—be it biased selection, drift, mutations, guided variation, or blending inheritance—mediate the observed relationships. These mechanisms' roles in influencing cultural traits' frequency and emergence are crucial for aligning the study more closely with significant cultural evolutionary frameworks and enhancing its impact in the field. The authors define "cultural diversity" in terms of variation in listening habits or music engagement. This differs from the traditional understanding, which encompasses variation in human cultural traits such as the structure and forms in music or language (Lomax, 1959; Rzeszutek et al., 2012). A more comprehensive explanation of how these two forms of diversity are related would clarify the study's contribution to cultural evolution.

We greatly value the thorough and constructive feedback from the reviewer.

We agree that our notion of cultural diversity is different from the typical one in the literature. Typical approaches to understanding diversity in music (e.g., Lomax, 1959; Rzeszutek et al., 2012; Mehr et al., 2019; Savage et al., 2015) examined diversity in *production* aspects of music, the musical content itself. Our approach for measuring within and between music diversity through *listening/consumption*, thus complements other studies by looking into an important but albeit less studied aspect of music consumption dynamics: the consumer habits. We now reflect this in the Introduction by describing different forms of diversity that exist, their importance, and also state explicitly our contribution:

L79: Including the work by Street et al., most research on music diversity has focused on the producer's perspective — analysing the varying types of musical artefacts that are produced (33,34,41,42). Nevertheless, both production and consumption, and their feedback loops with underlying selection pressure, are crucial for a comprehensive

understanding of the cultural evolutionary processes (43), particularly in the modern world of rapidly diffusing information and trend dynamics. Here, we focus on the consumer's perspective – analysing the varying ways in which individuals engage and listen to music (44,45).

Furthermore, we now extensively add to the Discussion, a comprehensive view of types of cultural diversity and how our findings might relate to some evolutionary mechanisms:

L412: *Our consumer-centric perspective on cultural diversity complements the producer-centric perspective that has been the dominant form in studies focusing on population size and culture (11,31,68). Previous studies of musical diversity have primarily focused on the content itself, such as the melody and rhythm of songs (33,34,69), considering these as the units of cultural traits. Our study, however, fills an important gap by analysing an often overlooked dimension: individuals' musical tastes as cultural traits that can evolve and is transmittable. Importantly, cultural variations within a population can take multiple forms, including differences in production (11), preference or consumption (32,44), and perception (37). A comprehensive understanding of cultural diversity necessitates considering all these aspects.*

Cultural traits resulting from production and consumption interact intimately through feedback cycles, which are common in complex systems, including culture and economy (43,70,71). For instance, a diverse range of consumption tastes might stimulate producers to create a broader array of cultural products, with social feedback signalling this demand. Conversely, individuals exposed to a wider range of cultural inputs, and consequently possessing a broader cultural breadth, may naturally tend to construct cultural niches. Hence, understanding listening behaviour can explicitly characterise how social selection pressure biases the creation of music. Besides, selection biases significantly influence the cultural production studied from historical records, as artefacts that were widely consumed are likely to have survived and thus documented (72). The dynamic interplay between production and consumption could reveal deeper insights into these co-evolutionary processes. Future studies might consider simultaneously measuring the diversity of cultural products and consumption, relating them to population size and demography.

Our findings also suggest the presence of other potential evolutionary mechanisms. Unlike genetic sequences, which primarily rely on vertical transmission (i.e., parents passing their genes to offspring), cultural evolutionary processes are thought to exhibit significant variations within the same generation (horizontal transmission) through social learning (73) – music sharing among friends for instance. Notably, an individual's cultural trait is not static like genes; one's musical taste and their breadth of exploration

can change even within their lifespan, as seen with our results of WID as a function of age in Fig. 2b.

The frequency of cultural traits, including music preferences, can shift over time due to a process known as 'cultural drift' (74). This drift has a pronounced effect in smaller populations, causing significant changes in trait frequencies. However, in larger populations, these frequencies tend to remain fairly stable. Our research supports this, as it shows a higher uniformity in music tastes in smaller areas, as indicated by a lower BID (Fig. 1b) and dense concentrations in user vector space (Fig. 1e). As a result, cultural diversity in large population areas may also be preserved for extended period of time, in turn influencing vertical transmission of cultural taste. Children growing up in large cities will be exposed to a broader spectrum of cultural content and have greater access to a variety of musical training programs, and this exposure can significantly influence their music perception (75). Our current data, which is a static snapshot, is unable to capture these potential vertical transmission mechanisms. Future research would benefit from examining longitudinal data to understand how individual and regional taste traits evolve over time.

References

- Lomax, A. Folk Song Style. *American Anthropologist* **61**, 927–954 (1959).
- Rzeszutek, T., Savage, P. E. & Brown, S. The structure of cross-cultural musical diversity. *Proceedings of the Royal Society B: Biological Sciences* **279**, 1606–1612 (2011).
- Mehr, S. A. *et al.* Universality and diversity in human song. *Science* **366**, (2019).
- Savage, P. E., Brown, S., Sakai, E. & Currie, T. E. Statistical universals reveal the structures and functions of human music. *Proceedings of the National Academy of Sciences* **112**, 8987–8992 (2015).

R1.2: The cultural breadth hypothesis posits that continuous exposure to diverse cultural inputs broadens one's cultural taste. Is this hypothesis supported by experimental evidence from psychological studies, or is it merely theoretical? A brief explanation with references to experimental findings would strengthen this section.

We thank the reviewer for this comment. Our work shows empirical support for the ‘cultural breadth hypothesis’, a hypothesis thus far mostly discussed as either theoretical propositions (e.g., cultural omnivore theory). To our knowledge there is no direct evidence from psychological studies or large data (like ours) thus far. We now make this explicit in the Introduction:

L69: Furthermore, while the link between socio-economic status and cultural breadth is well-studied (e.g., cultural omnivorousness (32)), there lacks empirical evidence on how the richness and size of one's cultural environment shape their cultural taste.

In addition, we add a statement in the Discussion about the empirical support we provide to the cultural breadth hypothesis:

L380: We add new insights using individual patterns: an individual's cultural breadth similarly expands with population size. This provides direct empirical evidence for the ‘cultural breadth’ hypothesis that thus far remained largely theoretical.

R1.3: “We collected data from France, Brazil, and Germany, which are the three countries with the largest percentage of Deezer users relative to their general populations.” How do these populations represent global urban versus rural dynamics? A discussion on their representativeness in the context of global urban-rural dynamics would be valuable.

To address this issue, we now conducted a new analysis to test the representativeness of these three countries in their urban versus rural populations by gathering additional data (Our World Data; <https://ourworldindata.org/grapher/urban-and-rural-population>; year 2022).

Analysis of this data revealed that, globally across the 229 countries, 62.0% (SD = 23.4%) on average live in urban areas. The three countries in our study, France (81.5%), Brazil (87.6%) and Germany (77.6%) were all above this average, and thus reflect characteristics of more urban societies. We now include these details in the Methods:

L555: Population census data showed that the majority of the population in these countries live in urban areas (France = 81.5%; Brazil = 87.6%; Germany = 77.6%), which were all above the world average (M = 62.0%, SD = 23.4%), and thus represent characteristics of highly urban countries.

and further elaborate as a Limitation:

L470: *Second, our study replicates the trends in different countries best covered on the Deezer platform, but the findings are limited to industrialised countries with the majority of the population living in urban centres (User demographic in Methods), and our causal inference testing is limited to France due to the spatial granularity of data.*

R1.4: While music is an ideal test-bed for examining cultural diversity due to its universal prevalence and cultural distinctiveness, potential drawbacks exist. What are the limitations of focusing solely on music? How might findings generalize to other cultural domains? Addressing these questions would provide a more balanced view of the study's scope.

We now add in the Limitations that acknowledge the limitations of only looking into music, their drawbacks, and replications across other domains as a future outlook:

L460: *There are several limitations to our study. First, our study is limited to music. Despite music being omnipresent in all human cultures (34), not all music is easily accessible or documented, particularly in cultures with strong oral traditions (76). This can create a selection bias in favour of music that is registered on globalised media platforms and subsequently studied. The global music market is also heavily influenced by accelerating globalisation and major music labels that dictate what is broadcasted. These influences are often complex and difficult to disentangle. Although similar issues may persist in other cultural studies (72), extending our research to areas like language, books, arts, societal values, and traditions would help to test the generalisability of the phenomenon.*

R1.5: I did not completely understand how BID and WID were calculated. Based on the information provided in the manuscript, it appears that 10,000 unique music streams were sampled for each bootstrap simulation from a subset of 100 random users. Is that correct? Or were the 100 users to be bootstrapped (Fig. 2b)? I am confused by the use of “respectively” in the Statistical Analysis section. This point should be made clear.

Thank you for spotting this. We want to first clarify that BID is computed using Hill's number (q order of 1, which becomes the Shannon entropy) across the unique *music streams* we sample from a given area. In contrast, WID is computed by sampling unique *individual users* and aggregating across their GS-Scores. We are using a fixed amount of streams (in BID) and users (in WID) because we want to avoid oversampling populated regions. Otherwise, the size of the sample itself can influence the diversity measure we compute. Thus, we randomly sample a fixed number, but repeated this process multiple 1,000 times so that different unique streams/users are selected per bootstrapped dataset.

It appears the reviewer's confusion stems from an error in our original manuscript of the sampling procedure, where the sequence of BID and WID sampling was mistakenly reversed. We now correct this in the Methods:

L801: *For BID, 1,000 unique streams were drawn from each area, while for WID, 100 unique individuals were drawn for each bootstrap. We sample a fixed amount of streams (BID) and a fixed amount of users (WID), so that larger areas are not a-priori over represented.*

Note that we now adjust our approach to sample 1,000 music streams for BID calculation, down from the initial 10,000. This is due to our decision to separate organic and algorithmic streams in response to concerns about the influence of algorithms (R1.9 and also R3.6 by Reviewer 3). We now focus only on organic streams for our main results of Figure 1&2 (but still include algorithm-not-excluded and using only-algorithm versions in SI). As a result of this new grouping, we chose a lower sampling number to accommodate smaller areas with limited data. Since we perform enough bootstraps, all results generally remain identical.

R1.6: In Fig 1c, do the significant correlations hold when music streams are grouped by artist and genre in control analyses? This information is not found in the supplementary material but should be included to ensure transparency and robustness of the findings.

Thank you for proposing this, which was also noted by Reviewer 3 (R3.4). We now include the stats when presenting in the main Results:

L161: *First, we expand our groupings of analysis and test beyond frequencies of 'songs', to 'artists' and 'musical genres' (Fig. 1c) and found these positive correlations to hold (grouping by artist: $r_s > 0.40$; grouping by genre: $r_s > 0.33$; all $p_s < .001$; Supplementary Table 1).*

R1.7: Supplementary Fig. 4 shows significant correlations between BID and population size for algorithmically recommended streams. Do these correlations persist when using alternative groupings, such as by users and genre? Including this analysis in the Supplementary Material would provide a more comprehensive understanding.

We now add a new Supplementary Table 2 (attached below) that shows the correlations across different grouping granularity, separating organic and algorithmic streams across all three countries. In general, segregating only organic, only algorithmic, and the two combined all revealed similar correlations and were all significant at $p < .001$ after correcting for multiple comparisons.

Table 2: Organic and algorithmic streams.

Pearson correlations between population size and BID computed independently for organic-only, algorithmic-only, and the two combined ('Algorithmic streams' section in Methods). 1,000 unique streams were drawn randomly per area and per listen type (i.e., organic and algorithmic listens). Generally, regardless of the grouping, the correlations were similar and all significant at $p < .001$ after adjusting for multiple comparisons. All values are derived from bootstrapped means and brackets indicate 95% CI ('Statistical analysis' section in Methods).

	Song			Artist			Genre		
	Organic	Algorithm	Combine	Organic	Algorithm	Combine	Organic	Algorithm	Combine
France	.73 [.62, .81]	.70 [.58, .79]	.75 [.64, .83]	.58 [.43, .70]	.63 [.49, .74]	.69 [.56, .78]	.41 [.23, .56]	.33 [.13, .49]	.42 [.24, .57]
Brazil	.40 [.28, .51]	.53 [.43, .62]	.48 [.37, .58]	.50 [.39, .59]	.63 [.54, .70]	.58 [.49, .66]	.33 [.21, .44]	.33 [.21, .44]	.36 [.24, .47]
Germany	.79 [.70, .85]	.73 [.64, .81]	.85 [.79, .90]	.40 [.23, .54]	.76 [.67, .83]	.65 [.52, .74]	.39 [.22, .54]	.42 [.26, .56]	.51 [.36, .63]

Considering the concerns about the influence of algorithmic bias (also noticed by Reviewer 3, R3.6), we now change the main analysis to only refer to organic (non algorithmic) streams, except in the case of causal testing (further elaborated in R1.9). We now note in the Method when describing BID:

***L625:** Given algorithmically recommended content can bias the calculation of BID, we excluded all algorithmic streams and used only the organic streams for the main analysis of Fig. 2 (Algorithmic streams in Methods). However, additional analyses that do not exclude algorithmic streams, and also in separation, can be found in Supplementary Table 2. For causal testing (Fig. 3), we did not exclude the algorithmic streams as we include this as a potential confounder (Algorithmic streams in Methods).*

Note that the correlation with genre is smaller. This may be because genre tags are noisy due to it relying on semantic categories. Genre tags are derived directly from the artist or the music labels themselves. Often a global genre tag of the artist can be uniformly applied across their songs, or set constant for the album — as such, individual song-level characteristics may get lost. Moreover, some may provide more specialised tags such as 'French rap', while others prefer to use more general tags like 'rap', where the two will be treated as unique categories when computing BID. We note these possible limitations also in the Supplementary Table 1 caption, which appears earlier and also uses the same groupings.

R1.8: The analysis is based on a four-week window of listening behavior. Discuss the temporal stability of these patterns and whether short-term trends or seasonal variations in music popularity could influence the observed BID levels. Acknowledging the limitations of this timeframe in capturing the full spectrum of temporal variability on BID levels, particularly in urban areas, is essential.

We agree that this is a limitation and now acknowledge this in the Limitations:

***L484:** Third, our data has a narrow temporal duration of a four-week window in March. This particular window was carefully selected because we wanted to minimise potential issues with people travelling during the summer holidays and Christmas periods, where mass population movement can bias our geographical-grounded data (77). However, it is possible that seasonal patterns at longer or shorter time constants can influence our results, and can be investigated in future works.*

R1.9: How do Deezer’s recommendation algorithms, which influence music exposure, affect the observed diversity metrics? Discuss the potential for algorithmic biases or personalized recommendations to skew BID, especially among heavy users who depend on these recommendations for music discovery.

We thank the reviewer for raising an important issue, which was also noticed by Reviewer 3 (R3.6). We now extensively revise the manuscript accordingly.

First, in Figure 1b, we re-ran the analysis using only organic streams to compute BID, while previously the streams mixed both algorithmically recommended and organic streams (original Figure 1b). As mentioned above, we include additional analysis for organic versus algorithmic streams separately, as well as the two combine as new Supplementary Table 2. We do this for all three countries and also group streams by songs, artists, and genres. We now add text in the Figure 1 caption to point the reader to this additional analysis:

***L159:** ... see Supplementary Table 2 for additional analyses including algorithmically recommended streams).*

Next, we also explored the question of whether the usage of algorithmic streams varies between small and large areas and include this as a new Supplementary Fig. 3 (attached below). We found significant negative correlations between algorithmically recommended stream proportion with increasing population size (except Germany). In other words, individuals living in large areas rely less on algorithmic recommendations and discover/listen to music more organically. To test if this phenomenon simply arises from population demography, we further controlled for age (Supplementary Fig. 3a) and gender (Supplementary Fig. 3b). Regardless of age group, the negative correlation persisted for France and Brazil, but not Germany. There were group-level differences in gender across all three countries — male users tend to use more algorithmic recommendations.

Fig. 3: Usage of algorithmic recommendations.

(a) Algorithmic recommendation proportion as a function of population size for different age groups. There is a negative correlation in France and Brazil, indicating that users in large

Now considering these variations across areas, we update Figure 3 (attached below) to include algorithmic recommendations as another confounder to the DAG and in causal testing. Here, we did not separate the organic and algorithmic streams (i.e. original version of BID) as the bias is now accounted for in the model, and that removing algorithmic streams will result in different number of stream contribution per individual and per area (thus becomes unbalanced). As for WID, the original computation already excludes the algorithmic streams as user's consumption breadth reflects only their explicitly chosen content. However, we still consider a potential bias and reflect it in the model, given the extent of algorithmic usage might also alter one's tendency to explore more or less of the content on their own. We note this in the Methods:

L741: *Given such apparent differences in algorithm usage by area size, we include the proportion of algorithmic streams as a confounder in casual testing. Although this algorithmic bias might affect our BID measure (Supplementary Table 2), it does not directly impact the WID since it is anyhow calculated solely based on users' explicit chosen content (i.e. organic streams). Nonetheless, an indirect pathway that could potentially influence the WID may exist, and we represent this in the DAGs (Fig. 3).*

and re-ran the entire model including the new variable and found nearly identical results.

Finally, we introduce a new section in the Methods dedicated to ‘Algorithmic streams’, that describes the potential biases introduced by the algorithmic recommendations:

L724: *Algorithmic streams*

Recent studies on the effect of algorithmic recommendations have shown a direct link to user’s diversity in music consumption, suggesting that individuals who tend to have more diverse taste rely less on algorithmic recommendations and engage in more organic exploration (49). To test for possible differences in algorithmic recommendations usage across population areas, we sub-sampled 10,000 individuals at random per area in our dataset and across the three countries. We then computed the proportion of their algorithm-driven music streams over all streams, sub-grouped by age and gender. Algorithmic streams include listens such as auto-played next song after listening to a song or an album, and personalised recommendations. Organic streams, on the other hand, are user’s explicitly chosen content such as searches and songs already included in their music library. In France and Brazil, there was a moderate to strong negative relationship between algorithmic stream proportion and population size across all age groups (see Supplementary Fig. 3 for statistics). This suggests that individuals living in large metropolitan areas tend to rely less on the use of algorithmic recommendations, however, the effect was small. There was also an effect of gender across all three

countries where male users tend to use algorithms more than female users. Given such apparent differences in algorithm usage by area size, we include the proportion of algorithmic streams as a confounder in casual testing. Although this algorithmic bias might affect our BID measure (Supplementary Table 2), it does not directly impact the WID since it is anyhow calculated solely based on users' explicit chosen content (i.e. organic streams). Nonetheless, an indirect pathway that could potentially influence the WID may exist, and we represent this in the DAGs (Fig. 3).

We also refer to the algorithmic bias more extensively within the main Results:

L166: *Third, we extensively test for regional differences in the usage of algorithmic recommendations and their influence on the observed outcome (Supplementary Tables 2 and Fig. 3; Algorithmic streams in Methods).*

R1.10: **In Figure 1D, only two regions (Paris and Orne) were used to measure regional user dispersion. Is there a reason for not including more samples to support this approach's validity? Including additional examples of both highly populated and smaller areas in the Supplementary Material would strengthen the analysis.**

Thank you for pointing this out (also noticed in R3.5 by reviewer 3). We think there was a misunderstanding. The two regions were only examples for illustration purposes, and we did not specify them specifically for the general analysis, which was computed across all regions. To avoid confusion, we now update Figure 1d (attached below) to show correlations with all regions and include the two areas examples as insets.

We revise the manuscript:

L188: *Using France as a case example, Fig. 1d demonstrates the strong correlation between the frequency-based BID measure and the amount of dispersion of users in the high-dimensional vector space ($r = 0.75$ [0.65, 0.83], $p < .001$). The dispersion amount also correlated highly with population size ($r = 0.54$ [0.38, 0.67], $p < .001$), validating the robustness of our results. Two areas are included as Fig. 1d insets as further illustrative examples to demonstrate the concentration of users in the vector space using kernel density estimation (see Supplementary Fig. 5 for additional examples).*

We also include new Supplementary Fig. 5 (attached below) as zoomed out version of the new Figure 1d where all French area labels are visible and more density map examples are included.

Fig. 5: BID and user dispersion.

Extended version of Figure 1d in the main text comparing the BID measure with user dispersion. User dispersion is computed based on the cosine distance across all pairs of sampled individuals in a given area (i.e. full matrix), and then computing population variance ('Dispersion

R1.11: Regarding the WID analysis by age, how much of the observed trend can be attributed to biases in the population structure? For instance, if the sample predominantly consists of participants in their 20s, with fewer older users (as indicated by larger error bars with increasing age), this potential bias should be discussed.

Thank you for pointing this potential issue, which was also noticed by Reviewer 2 (R2.1). The WID is computed from the embedding vectors of streamed music on the platform, in which this embedding is established through listening behaviour predominantly driven by younger users (as seen in skewed distribution of user age in Supplementary Fig. 9). This could indeed have contributed to the larger confidence intervals among older users.

We now acknowledge this as limitation, but also performed additional control analysis to complement the findings of Figure 2b, presented as new Supplementary Fig. 8 (attached below). We used diversity of *genres* consumed by each individual as an alternative proxy of WID. The idea is that the embedding that forms the basis of our WID is thought to be generally aligned with music genres (i.e., songs of similar musical styles tend to cluster together). The outcomes replicated and we reflect this in the main Results:

L260: *We also replicated the inverted U trend using an alternative approach by measuring one's diversity through their varied listening of musical genres (Supplementary Fig. 8). This validation was crucial considering the reliance of song embeddings to compute WID can inherently be biased due to the skewness of demography of Deezer users towards younger generation (Supplementary Fig. 9).*

Fig. 8: Within-individual genre diversity.

Within-individual level genre diversity as a function of age. Genre diversity per individual was computed based on the frequency of genre tags for the 100 streams sampled per user. Hill's number of order $q = 1$ was used for consistency with the BID measure ('Between-individual diversity (BID)' section of the main text). When the song contained multiple genre tags, the primary (the first) label was used. For each age category in bins of five years, 1,000 users were drawn at random, resulting in a total of 10,000 user samples per country. Among these sub-samples, bootstrap simulation was performed to compute the 95% CI. The result mirrors the general inverted U-shaped pattern of WID over age reported in Figure 2b of the main text.

R1.12: While the study controls for several socio-demographic variables, other potential confounders, such as media consumption habits or cultural norms, may influence the relationship between population size and diversity. Differences in internet connectivity and media exposure between urban and rural areas could also be factors. If obtaining this information from statistical institutes is not feasible, these limitations should be acknowledged.

We agree that our statement about missing confounders was not specific enough. For now, we updated the algorithmic biases as a confounder in the DAG model (updated Figure 3). Future work can also investigate ways to collect regional proxies or census data on the aforementioned factors suggested by the reviewer. We now note this in the Limitations:

L491: *Finally, we acknowledge that there may be other important confounders we did not consider, and that more demographic data could ultimately change the results. Specifically, different cultural norms and disparities among urban and rural areas in their internet connectivity and media exposure may be important factors to further consider*

R1.13: The study uses inverse propensity weights (IPW) to balance confounding variables across groups. How effective is this weighting method, and could there be potential issues with extreme weights or unbalanced covariates affecting the results?

Thank you for point this out. We would like to clarify that our procedure already takes into account extreme weights, and we do it in accordance with best practice in the literature. We now include the previously missing details regarding weight trimming in the new Supplementary Note 1.

Trimming/winsorizing propensity scores has been extensively studied in the literature as a solution to addressing extreme weights that might influence the outcome. However, finding the optimal cutoff threshold for weight trimming can be challenging (Lee et al., 2011) and whether such arbitrary criteria are necessary is currently a highly debated topic in the causal inference community at large. Newer methods such as the Average Treatment for the Overlap Population (ATO) weights come with a nice property that they yield an effect estimate with the smallest standard error of any weighted estimate, so the problems of instability are mitigated (Li et al., 2019). Weights are bounded at 0 and 1 so they are never too large or too small.

This ATO weighting procedure is what we apply in our study, which is less susceptible to the extreme weight problem. Therefore, we did not artificially trim the weights. We now explicitly describe this in the new Supplementary Note 1 with the relevant section copied below:

We applied the Average Treatment for the Overlap population (ATO) weighting procedure using the R package WeightIt6. Unlike traditional propensity score methods such as ATE or ATT, ATO weights mitigate issues of extreme weight variance as they are inherently bounded between 0 and 1, thus avoiding the need for trimming or winsorising (for issues related to weight trimming, see Lee et al. (7)). This was supported by our diagnostic checks which showed that our model achieves good covariate balance, demonstrated through standardised mean differences and empirical cumulative distribution function analyses (Supplementary Figs. 14 and 15).

References

- Lee, B. K., Lessler, J. & Stuart, E. A. Weight trimming and propensity score weighting. *PLoS one* **6**, e18174 (2011).
Li, F., Thomas, L. E. & Li, F. Addressing Extreme Propensity Scores via the Overlap Weights. *American Journal of Epidemiology* **188**, 250–257 (2019).

R1.14: The study focuses on France due to its high spatial granularity and demographic data. Discuss more thoroughly how generalizable the findings are to other countries or cultural contexts, particularly those with different population structures or music consumption patterns.

We agree this is a shortcoming of our study and now emphasise it better in the Limitation section with reference to new analysis we conducted in R1.3:

***L470:** Second, our study replicates the trends in different countries best covered on the Deezer platform, but the findings are limited to industrialised countries with the majority of the population living in urban centres (User demographic in Methods), and our causal inference testing is limited to France due to the spatial granularity of data. With the exception of age and gender, other demographic factors for causal testing were approximated from postcode level averages through inferred users' home locations. Thus, the regional data may be limited in fully capturing the subtle variances across the individuals. Moreover, cultural context are known to significantly influence music consumption patterns (39), and the user characteristics of the Deezer platform may not perfectly align with the general population's behaviour — although the general demographics appear to be similar (User demography in Methods). Future studies can incorporate wider global coverage to test if the results replicate in other countries, particularly focusing on ones with a larger proportion of rural populations.*

Reviewer 2

R2.1: The authors cannot show that age is the relevant variable explaining differences in BID/WID. In order to do that, they would need to have measurements over time. As of now, their results could be just a result of cohort (people born in the 50s having overall less diverse musical taste than people born in the 80s even conditioning on age). Please make this clear.

Thank you for pointing this out. We agree that WID as a function of age (Figure 2b) analysis cannot be fully disentangled from the era in which participants are born without a longitudinal evaluation of how one's musical taste evolves over a significant period. Such datasets for our paper would be hard to obtain given that online music streaming platforms have only been in existence for about a decade. We now acknowledge this as a limitation when presenting these results:

L266: It is important to note, however, that computed WID here reflects both the real age of the user as well as the period they were born in. Therefore, a cohort effect may exist whereby individuals born in a particular era have more or less diverse musical taste. Disentangling these associations would require a longitudinal data that tracks individual's change in music taste diversity over an extensive period of time. Future studies can incorporate datasets that track individuals' listening behaviour over many years (52).

R2.2: "...there was no discernible difference in the cultural breadth of individuals living in medium (inhabitant size around 500,000) versus large regions (more than 1.2 million), suggesting an upper limit in the extent that size fosters cultural exploration. This limit is likely reached when a region has sufficient infrastructure to support diverse cultural experiences." I think it's a bit premature to over-generalize any conclusions on the basis of three cities alone - it won't harm some nuance in these statements.

We now tone down the statement and suggest this as a potential explanation:

L404: ... there was no discernible difference in the cultural breadth of individuals living in medium (inhabitant size around 500,000) versus large areas (more than 1.2 million), suggesting a possible upper limit in the extent that size fosters cultural exploration. This limit could potentially arise from there being sufficient cultural infrastructures to support diverse cultural experiences once cities become sizeable enough, though this conclusion may require further replication in other countries to account for potential confounds resulting from other demographic factors.

We would like to clarify that the trend we find is not based on three cities alone, rather it is a group aggregate of many geographical regions (a total of 96 NUTS3 unit areas) that fall into the three small, medium, and large categories. We now clarify this in the text:

L307: *We first categorise user samples into three population quantile categories with each containing 32 areas ...*

Reviewer 3

R3.0: In general, the methods of this study appear sound and well justified, with careful consideration of potential confounds. We have, however, a number of queries and suggestions for the authors to consider concerning their methodology. We cannot necessarily comment on all of the aspects of the statistical methodology and accept that some of our questions may reflect our lack of familiarity with some of the specific techniques included in the paper. We are experienced in methods commonly used in cultural evolution and in some music information retrieval techniques, but some aspects of the methodology (particularly directed acyclic graphs) are less familiar to us. However, publications in such a broad interest journal should be as accessible as possible to researchers from a wide range of fields and so additional clarification on key aspects of the methods would be beneficial in any case. We appreciate that there is limited space in the main text to address all our comments in full, however, in some cases all that is required is rewording and/or clearer signposting to relevant areas of the Supplementary Information.

We greatly appreciate many valuable feedback from the reviewer. As suggested by the reviewer, we took this opportunity to extensively explain the causal inference procedure by including a step-by-step procedure section as new Supplementary Note 1. We then point the readers who are more interested in the details of the method to SI in the main text without making the main text lengthy:

***L307:** The step-by-step procedure we applied for causal testing is detailed in Supplementary Note 1, which generally adopts the protocols proposed by Ankan et al. (64).*

Here is the full new SI note copied below:

Note 1: Causal inference step-by-step procedure.

Our study employs a structured causal inference approach to explore the impacts of demographic and social factors on between and within-level diversity outcomes. Here, we describe the full step-by-step procedure of causal inference testing we applied, which generally adopts the protocol proposed by Ankan et al. (1).

1. Constructing the DAG

Using Directed Acyclic Graphs (DAGs), we modelled the hypothesised causal relationships among variables that guide our analysis. Potential confounders were chosen based on existing literature and they are detailed in the ‘Socio-demographics’ section of the main text. DAG provides a clear and efficient method to identify, present, and hypothesise the causal relationships between variables. Our main aim in illustrating

the model with a DAG is to transparently lay out our assumptions that can be refuted and corroborated by future researchers to serve as a useful framework.

2. Assumptions and sanity checks

Given that the causal testing we perform assumes normality, we performed log transformation for variables that did not follow a normal distribution as assessed by Q-Q plots (see Supplementary Table 5 for the distribution of all variables). Next, prior to model fitting, we assessed potential collinearity among the variables through covariance and correlational analyses. No pairwise correlations were larger than 0.92 and no variable was a linear combination of two variables.

3. Model specification

We defined two models for testing causal effects, as illustrated in Figures 3a,b in the main text:

Model 1: Effects of population size on between-individual diversity (BID)

Model 2: Effects of population size on within-individual diversity (WID)

For each model, we first tested the implied conditional independence of our DAG using the localTests function using the R package dagitty (2). Implied conditional independence is a method used to verify whether the assumptions of conditional independence hold true in a given probabilistic graphical model. In other words, it checks if some variables in the model are indeed independent of others, given the values of some other set of variables. The test is important because it helps ensure that the model accurately represents the relationships and dependencies among the variables. If the test fails (i.e., large effects are observed between variables that are not included as links in the model), it implies the model may need to be adjusted to more accurately reflect the true relationships among the variables. On our DAG, the test revealed no substantially large effects, suggesting the relationship we assume is not refuted by the data.

Next, we identified the minimal adjustment sets (i.e., variables to include as confounders). Minimal adjustment sets are the smallest sets of variables that, when controlled, can eliminate confounding bias in causal inference studies (3). Identifying specific confounders to control, rather than controlling for all possible confounders (i.e., throwing everything into the sink), is important to avoid misleading conclusions that can arise from relationships between the variables, such as the ‘collider bias’ (4).

4. Propensity score weighting

The minimal adjustment sets identified for each model in the previous section were controlled using the propensity score weighting method (5). This method involves estimating the probability of treatment assignment (in our case, the size of the area they live), known as the propensity score, for each individual based on their observed covariates. In a randomised experiment, this treatment probability is known, but in observational studies, it needs to be estimated, typically using logistic regression.

Once the propensity score is estimated, it is used to weigh each individual by the inverse of the probability of receiving the treatment they actually received. If they did not receive the treatment, they are weighted by the inverse of one minus their propensity score. This process synthesises a sample in which the distribution of observed covariates is independent of treatment assignment. In essence, this method summarises all covariate information into a single score for each participant, which is then used to create a balanced and unbiased sample for estimating the causal effect of the treatment. This mimics the scenario of a randomised controlled trial, thereby reducing the bias due to confounding variables, and allowing for more accurate estimates of causal effects in observational studies.

We applied the Average Treatment for the Overlap population (ATO) weighting procedure using the R package WeightIt (6). Unlike traditional propensity score methods such as ATE or ATT, ATO weights mitigate issues of extreme weight variance as they are inherently bounded between 0 and 1, thus avoiding the need for trimming or winsorising (for issues related to weight trimming, see Lee et al. (7)). This was supported by our diagnostic checks which showed that our model achieves good covariate balance, demonstrated through standardised mean differences and empirical cumulative distribution function analyses (Supplementary Figs. 14 and 15).

The obtained propensity scores for each individual were then used as weights in a linear model when predicting the outcomes of BID and WID as a function of population size.

5. Error estimation

To assess the robustness of our models, we used bootstrap sampling with replacement to obtain confidence estimates. We ran 1,000 simulations each with newly sampled sets of individuals and by reapplying the entire weight assignment procedure.

R3.1: The cultural diversity measures could be more clearly explained, particularly for the benefit of broad audiences. The between-individual diversity (BID) measure has been adapted from species diversity measures used in ecology, and a little more explanation of how it translates to cultural diversity would be helpful. For example, if songs are analogous to species (L476), what is the cultural equivalent of an ecosystem – a geographic region?

We now provide a clearer intuition in what BID measures in the main Results:

***L126:** As a measure for between-individual diversity (BID), we adopt Hill's number (with order of 1, which effectively becomes the Shannon entropy; Measuring diversity in Methods), a widely used method for measuring cultural diversity that allows for standardised comparisons (46–48). A high overlap of songs among individuals indicates a more shared consumption of musical repertoire, resulting in a greater skewness in the frequency distribution, where consumption is concentrated on a highly popular set of songs (i.e., low diversity, low BID). Conversely, a low overlap indicates there being a greater distinction between individuals, resulting in a flatter frequency distribution, where consumption is more distributed (i.e., high diversity, high BID).*

and then further clarify the measure with direct references to ecology in the Methods:

***L602:** Originating from ecology, Hill's number treats the abundance of species in an ecosystem (or field site). In our use case, we treat each song as a species and the abundance of songs streamed in a geographical area as their ecosystem (i.e., NUTS3 areas of France and Germany, municipalities of Brazil). It essentially quantifies how diverse the music consumption is in a given area, indicating whether individuals listen to a wider range of songs or concentrate on a few popular ones.*

R3.2: Similarly, the within-individual diversity (WID) measure could be explained in a more user-friendly fashion – phrases like ‘general song embedding space’ (L509) are not likely to be intelligible to broad audiences.

We now extensively revise throughout the paper to use a more general reader-friendly term when possible (e.g., high-dimensional vector space) and provide clear explanations in how the WID is computed and what it captures.

First, we provide a more detailed and intuitive explanations about what WID captures in the main Results:

***L209:** To assess within-individual diversity (WID), we adopt a metric known as the Generalist-Specialist Score (GS-Score) of music engagement (45,49) by tracing each individual's listening history (Measuring diversity in Methods). Here, we again utilise a*

high-dimensional vector space, but this time, capturing the relationships between songs, created based on millions of user interactions with songs on the Deezer platform (Song embedding in Methods). Next, we calculate each user's 'coverage' in this vector space by mapping all the songs they have listened to in the past 28 days. This coverage, or the radius in the vector space, is used as an indicator of their overall breadth of music engagement. If a user is a 'specialist', their listening habits will be concentrated in a specific location of the vector space (for example, they may only listen to K-pop), and their radius will be small. Conversely, a 'generalist' user will have a wider coverage, indicating they listen to a diverse range of music (for example, they may listen to K-pop, metal, and classical music). To compute this radius, we use the average cosine similarity between a randomly selected song and the average of all songs the user has listened to (Measuring diversity in Methods).

Second, we also improve on clarity when describing WID in the Methods :

L634: *The GS-Score computation relies on the high-quality song embeddings of Deezer that summarises relationships between songs as high-dimensional vector representation (Song embedding in Methods). First, a user's (u) centroid position in the song embedding is defined by computing the mean vector ($\vec{\mu}$) of all the songs the user has listened to within the last 28 days. Next, the cosine similarity is calculated between $\vec{\mu}$ and a randomly selected song (\vec{s}) the user listened to, weighted by the number of times they have listened (w_s). This approach ensures the measure is not sensitive to the number of songs that the user has listened to. Moreover, WID is computed based solely on user's explicitly chosen content, thus do not include algorithmically recommended streams. The resulting GS-Score ultimately captures the user's radius of coverage in the song embedding, formally written as:*

Last, early on in the Methods, we provide a description of what 'embedding' is for those who may not be familiar:

L573: *Many recommendation systems leverage embeddings to efficiently encode the latent relationships between users and content. Embedding is a mathematical space where objects, such as songs in this context, are represented as vectors in a way that captures the relationships among them. In this space, closely related songs are positioned near each other. For instance, tracks that share thematic or stylistic similarities, such as Adele's 'Someone Like You' and Sam Smith's 'Stay With Me', would be located in proximity within this space.*

R3.3: As the authors note (L532-533), one important limitation of the BID measure is that it captures population-level diversity in the songs that are listened to, but not other dimensions of diversity (e.g. genre/stylistic diversity) which are potentially more meaningful. The ‘dispersion in user embedding’ measure addresses this issue but as before it could be more clearly explained – the relevant section of the Methods (L514-L543) is technically dense and challenging to understand (even for those with some experience in music information retrieval). It is also not completely clear to us whether this issue also affects the within-individual diversity measure.

Thank you for these suggestions. First, we now include different groupings of BID by artist and genre and report these statistics in the main Results (also suggested by Reviewer 1 (R1.6)):

***L161:** First, we expand our groupings of analysis and test beyond frequencies of ‘songs’, to ‘artists’ and ‘musical genres’ (Fig. 1c) and found these positive correlations to hold (grouping by artist: $r_s > 0.40$; grouping by genre: $r_s > 0.33$; all $p_s < .001$; Supplementary Table 1).*

Second, we provide a more intuitive explanation of the dispersion in user embedding in the Results:

***L180:** We leverage a high-dimensional vector space that summarises the relations between users based on the similarity of their listening behaviour, where users who listen to similar music are situated closer in this space (e.g., both are K-pop fans), while far apart if they have distinguished interests (e.g., K-pop versus metal fan; User embedding in Methods). The amount of variance in the pairwise distance among the users is then used to capture the general homogeneity or heterogeneity in music interest of the local area (Dispersion in user embedding in Methods).*

and also in the Methods:

***L656:** An alternative approach that can provide a higher-level characterisation is to measure how misaligned users are in their music preferences by utilising distances within the user vector space (User embedding in Methods). As described above, users with similar musical preferences, or taste, will have a shorter distance within the space while far apart if they are distinguished. At the geographical area level, we can measure the pairwise distance across all users and measure the dispersion, or the radius, as indicator of diversity of music preferences of the given area. Formally, this user dispersion is computed by taking the population variance in the pairwise distances of users from a given area, written as:*

Last, WID computation is analogous to the dispersion in user embedding, but rather focusing on each individual's radius or breath, and thus does not rely on categorical grouping. We now make this point also clearer:

***L650:** As such, unlike BID, WID does not rely on categorical grouping and captures the distances within the vector space.*

R3.4: Readers can find some comparisons of song-, artist- and genre-level measures of BID in the SI but these are only very briefly referred to in the main text (L134). It appears that effects of population size are weaker for genre- than song-based BID measures, but only the strongest correlations are reported in the main text (L133) – the authors should be more transparent about this.

As responded in the comment above, we now report BID correlations using artist and genre-level grouping at the forefront in the main text. We reported song-level measure as it offers the highest granularity, however we also acknowledge that higher-level categories such as using genre indeed reveal weaker correlations.

Particularly the correlation with genre is weak, and we think this is due to genre annotations being noisy. Genre tags are derived directly from the artist or the music labels themselves. Often a global genre tag of the artist can be uniformly applied across their songs, or set constant for the album — as such, individual song-level characteristics may get lost. Moreover, some may provide more specialised tags such as 'French rap', while others prefer to use more general tags like 'rap', where the two will be treated as unique categories when computing BID.

To acknowledge this, we add the following text in Supplementary Table 1 (attached below) caption:

Note that the 'genre' grouping of BID reveals the weakest correlations. This may be due to genre tags being noisy. Genre tags are derived directly from the artist or the music labels themselves. Often a global genre tag of the artist can be uniformly applied across their songs, or fixed for the album — as such, individual song-level characteristics may get lost. Moreover, some may provide more specialised tags such as 'French rap', while others prefer to use more general tags like 'rap', where the two will be treated as unique categories when computing BID. These inconsistencies likely affect the results.

Table 1: Pearson and Spearman correlations.

Pearson and Spearman correlations with BID and WID. BID is separately computed by grouping the streams in the units of songs, artists, and genres. WID is measured using the inverse of GS-Score. BID is measured using Hill's effective number with q order of 1 (i.e., Shannon entropy; 'Measuring diversity' section in Methods). All values are derived from bootstrapped means and brackets indicate 95% CI ('Statistical analysis' section in Methods). Note that the 'genre' grouping of BID reveals the weakest correlations. This may be due to genre tags being noisy. Genre tags are derived directly from the artist or the music labels themselves. Often a global genre tag of the artist can be uniformly applied across their songs, or fixed for the album — as such, individual song-level characteristics may get lost. Moreover, some may provide more specialised tags such as 'French rap', while others prefer to use more general tags like 'rap', where the two will be treated as unique categories when computing BID. These inconsistencies likely affect the results.

	WID		BID					
	GS-Score		Song		Artist		Genre	
	Pearson (r)	Spearman (ρ)	Pearson (r)	Spearman (ρ)	Pearson (r)	Spearman (ρ)	Pearson (r)	Spearman (ρ)
France	.65 [.52, .75]	.60 [.45, .72]	.73 [.62, .81]	.74 [.63, .82]	.58 [.43, .70]	.54 [.37, .67]	.41 [.23, .56]	.37 [.18, .54]
Brazil	.32 [.20, .44]	.31 [.18, .43]	.40 [.28, .51]	.39 [.27, .50]	.50 [.39, .59]	.48 [.37, .58]	.33 [.21, .44]	.34 [.21, .45]
Germany	.34 [.17, .49]	.28 [.09, .44]	.79 [.70, .85]	.75 [.66, .83]	.40 [.23, .54]	.38 [.20, .53]	.39 [.22, .54]	.42 [.25, .56]

R3.5: Finally, it's not clear to us why the results are repeated using the dispersion in user embedding measure only in France, with results displayed only for two selected French regions in Figure 1. Instead, we suggest that correlations with all diversity measures are reported for all three regions in SI Table 1 so that readers get a full picture of the results and can easily compare strength of effects for different diversity measures.

We are sorry about this confusion. The two regions in Figure 1d were only examples for illustration purposes. It seems that this was a source of confusion also for Reviewer 1 (R1.10). To avoid confusion, we now update Figure 1d (attached below) to show correlations with all regions and include the two areas examples as insets.

Accordingly we revise the presentation of results:

L188: *Using France as a case example, Fig. 1d demonstrates the strong correlation between the frequency-based BID measure and the amount of dispersion of users in the high-dimensional vector space ($r = 0.75$ [0.65, 0.83], $p < .001$). The dispersion amount also correlated highly with population size ($r = 0.54$ [0.38, 0.67], $p < .001$), validating the robustness of our results. Two areas are included as Fig. 1d insets as further illustrative examples to demonstrate the concentration of users in the vector space using kernel density estimation (see Supplementary Fig. 5 for additional examples).*

We also include more density map examples in new Supplementary Fig. 5 (attached below):

Fig. 5: BID and user dispersion.

Extended version of Figure 1d in the main text comparing the BID measure with user dispersion.

User dispersion is computed based on the cosine distance across all pairs of sampled individuals in a given area (i.e. full matrix), and then computing population variance ('Dispersion

We agree that user dispersion also for Brazil and Germany would be good to add. But here we face practical limitations. We initially did not collect the raw embedding vectors for users outside France during the sampling because these require a lot of space (and some computation). Note that our procedure included a large data download of all relevant information (raw data was over 60G), which we provide as aggregate form in the OSF repository for further analysis. As such, changes to this initial pipeline would require recollection of all the data again, which we believe is outside the scope of this project. Thus, we clarify this limitation in the Supplementary Fig. 5 caption:

Note the analysis could not be run for Brazil and Germany as the raw user embeddings for these countries were not extracted during the data collection phase.

R3.6: The premise of the study, as established in the introduction, is that cultural diversity is the result of diversity between individuals in their experiences and/or within individuals in their preferences. However, hypotheses are tested using measures of cultural diversity derived from user behaviour on a music streaming platform, which is to a significant extent influenced not only by users' own predilections but by the choices made available to them by algorithmic recommendations. The authors do mention this issue but only very briefly (L425, 149). I think given the importance of recommender algorithms in user behaviour, it would be worth clarifying this issue further. It's particularly important to consider whether algorithmic recommendations may bias the effects – for example, do people streaming music in larger metropolitan areas receive more diverse algorithmic recommendations? In the Supplementary Information, the authors present a comparison in the relationship of a single measure of BID to population size between 'organic' and 'algorithmic' streams which shows qualitatively comparable results although effects are a little stronger when algorithmic streams are included, which suggests that they may introduce some biases in favour of the hypothesis. We suggest that the authors either 1) remove algorithmic streams from all of their analyses or 2) include a side-by-side comparison of all key results between organic and algorithmic samples in the Supplementary Material.

We thank the reviewer for this suggestion, which was also noticed by Reviewer 1. We decided to accept both of your proposals and provided a full response regarding algorithmic influences above (see detailed response to R1.9).

Importantly, as suggested by the reviewer, we decided to include only the organic streams for the main analysis of Figure 1&2, but still include algorithmic streams and combined version for comparisons (new Supplementary Table 2).

We also explored regional differences in the extent of algorithmic recommendation usage in the new Supplementary Fig.3, which revealed a significant relationship with population size. Therefore, rather than excluding algorithmic streams all together from the DAG, we decided to include algorithmic bias (i.e. regional proportion of algorithmic recommendation use) as an additional confounder in the model, which revealed nearly identical outcome (updated Figure 3).

R3.7: The observations in the analyses are not necessarily independent due to spatial effects: neighbouring regions are probably more similar in cultural diversity/population size than distant regions. Do the authors need to consider and control for spatial autocorrelation?

We thank the reviewer for this suggestion. To address this, we performed several analyses and include as new Supplementary Table 4:

We began by analysing the geographical spatial clustering of BID/WID to see if there are meaningful spatial correlations. We report the standard measure of Moran's I, where values closer to 1 indicate more spatial clustering while closer to 0 indicate more random distribution across the geography. As expected, Moran's I was significant in most cases, suggesting there is spatial clustering for the diversity measures, with the exception in the case of BID in Germany ($p = 0.32$). This indicates that, as the reviewers mentioned, there is generally spatial autocorrelation in the diversity measure we calculated for all three countries, with France showing particularly strong autocorrelation.

	Moran's I			
	BID		WID	
	Moran's I	p	Moran's I	p
France	0.50	< .001	0.50	< .001
Brazil	0.30	< .001	0.17	.003
Germany	0.04	0.32	0.42	< .001

Based on that, as additional control analysis, we performed spatial detrending by subtracting the mean values of neighbouring regions (excluding the focal region itself) from the dependent variable (either BID or WID) of the region. Neighbours were defined as regions within the 100km radius. This process ensures that spatial correlations are significantly reduced by removing the regional mean.

	Spatial detrending			
	BID		WID	
	Pearson r	p	Pearson r	p
France	.27 [.08, .45]	.007	.37 [.18, .53]	.001
Brazil	.60 [.51, .68]	< .001	.37 [.25, .48]	< .001
Germany	.75 [.66, .82]	< .001	.39 [.22, .54]	< .001

Note however that the minor changes in Brazil and Germany may be attributable to the considerable distances between their urban centres.

Finally, we also tested if the correlations of BID and WID still hold when computing areas with a lower spatial resolution. The idea is that if our results are simply artefacts of spatial correlation, then decreasing the resolution should substantially reduce the effect. In France and Germany, there exists NUTS2 region data that is a higher level regional boundary to NUTS3 used in our study. We thus tested for these two countries, while Brazil was excluded from this analysis due to not having equivalent multiresolution regional data. With BID, we found that the correlation is

still high in France and Germany (France: NUTS2 $r = 0.82$, NUTS3 $r = 0.73$; Germany NUTS2 $r = 0.45$, NUTS3 $r = 0.79$). In France, WID was slightly reduced (NUTS2 $r = 0.56$, NUTS3 $r = 0.65$) and in Germany the correlation that was already low diminished (NUTS2 $r = 0.04$, NUTS3 $r = 0.34$). This suggests that indeed in Germany some of the effects of increased WID with population size have to do with spatial correlation.

	Lower spatial resolution (NUTS2 level)			
	BID		WID	
	Pearson r	p	Pearson r	p
France	.82 [.60, .92]	< .001	.56 [.18, .79]	.007
Germany	.45 [.13, .69]	.008	.04 [-.30 .38]	.812

These results are now reflected in the main text:

L171: *Finally, we test the extent to which spatial auto-correlation is present and test for spatial effects through detrending analysis and broader geographical groupings (Supplementary Table 4). We observed consistent patterns across all of these cases: BID and local population size were highly correlated even when these confounders were tested empirically.*

Note however that our study primarily investigates the link between cultural diversity and population size, considering that urban proximity could be a contributing factor to diversity. In this context, we perceive the inherent spatial effects as an integral part of our analysis instead of a confounding variable that requires control. We also explain this issue in the Table caption:

Nevertheless, we note that our analysis focuses on the relationship between cultural diversity and population size, and proximity to an urban centre can be one of the reasons for diversity. In this respect, we view the existing spatial effects as a feature of our analysis rather than a confounding factor that needs to be controlled for. The phenomenon that large cities—and by extension, their neighbouring areas that are also usually large—exhibit more cultural diversity is a central aspect of our investigation.

R3.8: In the introduction, the two hypotheses are presented as separate, but it seems that the second depends on the first – even if people living in urban areas are more open to diverse cultural activities, without a ‘source’ of cultural diversity (i.e. individuals from diverse backgrounds, sharing diverse cultural ‘products’), then diversity surely cannot increase with population size.

We agree with the reviewer that the two hypotheses are very likely interconnected and worth mentioning in the Introduction. As suggested, residing in an environment with a high degree of interaction with individuals from diverse cultural backgrounds may facilitate the discovery and exploration of unfamiliar cultural products, thereby enhancing individual-level cultural diversity (i.e., WID).

However, we cannot fully agree that this is a *necessity* for cultural diversity to grow. As demonstrated by theoretical models of population growth and cultural diversity (Henrich, 2004; Derex et al., 2013), population size and increased social connectivity can independently drive cultural complexity, without requiring external influences (e.g., new cultural products introduced by migrants).

In response to this feedback, we now revise the Introduction:

***L61:** These two mechanisms could be interconnected. For instance, demographic mixing could create opportunities for interaction and exchange with individuals from more diverse cultural backgrounds, in turn serving as a catalyst for discovering new cultural products. Alternatively, sheer size alone could independently drive cultural diversity, as larger populations are more effective in preserving cumulative cultural knowledge^{11,31}.*

References

- Henrich, J. Demography and Cultural Evolution: How Adaptive Cultural Processes Can Produce Maladaptive Losses—The Tasmanian Case. *American Antiquity* **69**, 197–214 (2004).
- Derex, M., Beugin, M.-P., Godelle, B. & Raymond, M. Experimental evidence for the influence of group size on cultural complexity. *Nature* **503**, 389–391 (2013).

R3.9: In the data availability statement it appears that variables are maintained in separate datasets (i.e. github repository, census data, Songkick database, etc.). This makes it more challenging for readers to replicate the analyses presented in the paper – we would suggest that variables are combined into single datasets as much as possible.

We agree. The dataset is now combined into a single dataset that includes all census and other regional information gathered (repository updated: <https://osf.io/6zugm/>). The sources of the different datasets are indicated in the readme file of the repository.

The ‘Data availability’ section is now edited to provide a single link:

L811: *Aggregated data of BID and WID measures and all census data used in the study is available at <https://osf.io/6zugm>*

R3.10: We assume that Deezer streamers have consented to their data being included in research studies like this one through user agreements, but it would be good to be explicit about this.

We appreciate the reviewer's concern regarding the use of Deezer user data in our study. Indeed, as part of the user agreement, users have consented to their data being used for advancing scientific research, in line with the terms of service for *fair data use*. Note that our study does not look into individual-level data that could potentially be used to identify specific individuals. To ensure transparency, we now clarify this point as follows:

L526: *The data was handed to the researchers anonymised and the analysis was not used to derive commercial profiling of any kind. In accordance with the Deezer user agreement, users have consented to their data being used for the purpose of advancing scientific research.*

R3.11: The section on casual inference (L618-635) is tough going for general readers – many key concepts (e.g. ‘implied independence’) are likely to be unfamiliar. It’s also not clear to us how the covariates were ‘condensed’ into a single ‘propensity score’. It may, further, be helpful to explain some of the limitations of causal inference for observational data. Could further detail and explanation be provided in an extended methods section in the SI?

Thank you for noticing this. The same issue was also raised by Reviewer 1 (R1.13). We now revise the paragraph introducing the DAG in the Results for more details to assist general readership:

L322: *Importantly, DAG allows the identification of which confounder should be controlled or left uncontrolled, rather than controlling for every imaginable covariates. Using this DAG, we test the direct effect of population size on BID (Fig. 3a) and WID (Fig. 3b) after controlling for demographic variables. We checked that several key assumptions are met, including implied conditional independence that test for whether the graph we assume is not refuted by the observed data (Causal inference in Methods). The step-by-step procedure we applied for causal testing is detailed in Supplementary Note 1, which generally adopts the protocols proposed by Ankan et al.64*

The new Supplementary Note 1 now describes the causal inference testing, which also describes implied independence and covers in-depth how covariates are treated using propensity score. Here is the relevant section on implied independence:

For each model, we first tested the implied conditional independence of our DAG using the `localTests` function using the R package `dagitty` (2). Implied conditional independence is a method used to verify whether the assumptions of conditional independence hold true in a given probabilistic graphical model. In other words, it checks if some variables in the model are indeed independent of others, given the values of some other set of variables. The test is important because it helps ensure that the model accurately represents the relationships and dependencies among the variables. If the test fails (i.e., large effects are observed between variables that are not included as links in the model), it implies the model may need to be adjusted to more accurately reflect the true relationships among the variables. On our DAG, the test revealed no substantially large effects, suggesting the relationship we assume is not refuted by the data.

Here is the relevant section on propensity score:

4. Propensity score weighting

The minimal adjustment sets identified for each model in the previous section were controlled using the propensity score weighting method (5). This method involves estimating the probability of treatment assignment (in our case, the size of the area they live), known as the propensity score, for each individual based on their observed covariates. In a randomised experiment, this treatment probability is known, but in observational studies, it needs to be estimated, typically using logistic regression.

Once the propensity score is estimated, it is used to weigh each individual by the inverse of the probability of receiving the treatment they actually received. If they did not receive the treatment, they are weighted by the inverse of one minus their propensity score. This process synthesises a sample in which the distribution of observed covariates is independent of treatment assignment. In essence, this method summarises all covariate information into a single score for each participant, which is then used to create a balanced and unbiased sample for estimating the causal effect of the treatment. This mimics the scenario of a randomised controlled trial, thereby reducing the bias due to confounding variables, and allowing for more accurate estimates of causal effects in observational studies.

R3.12: Results are interpreted in terms of urban versus rural differences (e.g. L134-136), while strictly they are effects of population size (which obviously differs between urban and rural areas, but population size is only a proxy for urban/rural differences).

We agree and now edit the text:

L147: This indicates that individuals living in more populated areas tend to have more distinct listening patterns from one another.

R3.13: The authors describe ‘cross-cultural differences’ in diversity (L139) but they are referring to differences between three large, multicultural nations (France, Germany and Brazil) so perhaps ‘between-country differences’ would be more appropriate. It is also not clear whether these comparisons account for socio-demographic differences between the countries.

We agree with the reviewer that the term ‘between-country difference’ is more specific and appropriate. Here are relevant lines we revised:

L149: Interestingly, our analysis also revealed notable between-country differences ...

L235: While we previously saw substantial between-country differences in the level of BID ...

The comparison between countries does not take into account the socio-demographic differences. This is to be consistent with the raw correlation reported for BID or WID in Figures 1a & 2a, which do not take into account the demographics. It is only in Figure 3 with causal inference testing, that extensive demographic details are taken into account.

R3.14: Minor errors.

Here we provide a quick response, pointing to the corrected line in the revised manuscript.

L56: should ‘megapolis’ be ‘megapolises’?

Corrected:

L57: ...as seen in megapolises like Paris, Berlin, and São Paulo

L68-69: Street et al. (2022) found that tunes played by larger communities have more varied versions, but these tunes tend towards intermediate (not higher) complexity.

Corrected:

L77: ...Street et al. showed that tunes played by a larger community of musicians evolve into more varied versions and exhibit intermediate levels of melodic complexity.

L127: ‘resulting as’ should be ‘resulting in’

Corrected:

L159: ... resulting in dispersed pockets of diverse music interests.

L146: the term ‘control experiments’ may be confusing for readers as all analyses are based on observational data. Perhaps ‘additional analyses’ or similar would suffice here.
We now use the term ‘control analysis’:

L161: To examine the robustness of our results, we conduct a series of control analyses ...

L387: We confirmed the robustness of our results through a series of control analyses ...

L158-159: the phrase ‘dispersion amount of regional users in each region’ is challenging to understand.

Corrected:

L183: The amount of variance in the pairwise distance among the users is then used to capture the general homogeneity or heterogeneity in music interest of the local area (Dispersion in user embedding in Methods).

Using France as a case example, Fig. 1d demonstrates the strong correlation between the frequency-based BID measure and the amount of dispersion of users in the high-dimensional vector space ($r = 0.75 [0.65, 0.83]$, $p < .001$). The dispersion amount also correlated highly with population size ($r = 0.54 [0.38, 0.67]$, $p < .001$), validating the robustness of our results.

L376: typo (‘musicl’), also the Street et al. study looked at melodic rather than stylistic diversity

We now exclude this in the Discussion.

L450: ‘this resulted as’ should be ‘this resulted in’

Corrected:

L548: This resulted in 96 and 113 NUTS3 unit regions...

L478: ‘specie’ should be ‘species’

Corrected:

L603:...each song as a species and the abundance of...

L483: the ‘q’ parameter in Hill’s number could be more clearly explained – is it assumed a priori or estimated from the data?

The q is assumed a priori and is one that is commonly used as it is analogous to the measure of Shannon entropy (described in Equation 2). Clarified:

L622: *In our analysis, we set a priori the order of q to be 1, but results were also robust to other values of order q (Supplementary Table 3).*

L551-552: it would be interesting to know what the expected differences are in streaming behaviour between genders

Corrected:

L677: *Studies have shown how one's musical exploration (45) and preferences (51) are influenced by age, demonstrating that one's music taste generally consolidates during adolescence. Research has also shown significant differences between the consumption patterns of males and females, noting that male users on online platforms tend to consume more diverse and niche content (44,90).*

L557: 'self-identified themselves' is redundant, you can just say 'self-identified as'

Corrected:

L686: *...and who self-identified as Male or Female*

L558: the exclusion of participants over 65 and those identifying as other than male/female could be better justified

Clarified:

L687: *This was due to reduce noise as there were only small number of individuals per area outside of this criteria.*

L594-607: measures of social connectedness are based on Facebook data, but we wonder if this matches well with the streaming user demographics, especially given that Facebook use is declining among younger demographics

We now include additional analysis comparing the demography of Facebook user data by age and gender with users on Deezer. Facebook does not publicly release country-level demographic data. However, private marketing companies use the developer API systems to collect data about various demographic information. From the source drawn from a marketing company (<https://napoleoncat.com>), for the same month as our study window, we compared the proportion of users by gender per each age category and included as new Supplementary Table 6.

Table 6. France Facebook users demography comparison.

As a validation step for the 'social connection' measure, which we included as a confounder in the causal inference ('Causal inference' in Methods), we compared user demographics between Facebook and Deezer in France. We obtained demographic information for French Facebook users in March 2023 from a private marketing company (<https://napoleoncat.com/stats/facebook-users-in-france/2023/03>). We then used the age categories in this data to compare with our Deezer user sample.

Age category	Facebook		Deezer	
	Male (%)	Female (%)	Male (%)	Female (%)
13-17	1.5	1.7	3.5	5.6
18-24	10.0	10.4	16.0	15.8
25-34	12.2	12.4	9.3	11.6
35-44	8.9	9.8	8.2	9.8
45-54	6.7	7.6	5.8	7.8
55-64	4.3	5.6	1.8	3.0
65+	3.7	5.2	0.6	1.1

We also add the following paragraph in the relevant section:

L769: *To validate how Facebook user demography compares with the Deezer users in France, we collected data from a private company (<https://napoleoncat.com>) that gathers country-level Facebook user demographics, for the same month as our sampling window in March 2023. Demographics were similar for both females and males across the age groups (same Spearman correlations for both genders = 0.86 [0.29, 0.98], $p = 0.08$), with adolescents taking up the largest proportion in both platforms (see Supplementary Table 5 for raw values).*

L643: what method of adjusting for multiple comparisons was used?

Thanks for noting this missing information. The correction was applied using Holm approach and now clarified:

L805: *... coefficients were adjusted for multiple comparisons using the Holm method (101).*

Response to the Reviewers

Sep 19, 2024

Dear Reviewers,

We are delighted to hear that our substantially revised manuscript entitled “*Mechanisms of cultural diversity through 2.5 million individuals’ music listening patterns*” resolved all the major concerns, and we appreciate the further minor comments.

We have fully addressed all the issues raised by the reviewers and revised the manuscript as provided in a detailed response below.

To facilitate an easier task for the reviewers, we labelled each point raised by the reviewers and their comments as **bold** text. Our response is placed just under as normal text and reference to changes in the revised manuscript as *italic* text including line numbers in brackets. In the revised manuscript, all changes are highlighted in yellow. In addition, we provide a clean version with all changes resolved.

Considering the comment from Reviewer 3 regarding the paper's title, we now propose a more concise but comprehensive title to appeal to a broader audience: “*Why cities vary in their cultural diversity*”.

After resolving these issues, we believe the manuscript now improved in its clarity and depth.

Sincerely,

Harin Lee, Nori Jacoby, Romain Hennequin, Manuel Moussallam

Reviewer 1

R1.1: first paragraph in the Discussion section: Clarify again what aspect of ‘cultural diversity’ the study addresses (i.e., music consumption). This may be useful for readers who only skim through the key structural points of the article, such as the abstract and the first paragraph of the discussion, to assess whether the paper aligns with their interests.

We appreciate this suggestion. We now mention this aspect in the Discussion:

***L376:** Using music listening patterns in the real world, we sought to decipher this and study the mechanisms and factors contributing to cultural diversity, here defined as diversity in music consumption.*

**R1.2: "[...] We observed an increasing trend of between-level diversity [...]"
Between-individual diversity?**

Thank you, we corrected the mistake:

***L382:** ... we observed an increasing trend of between-individual diversity with the population size of the area ...*

R1.3: "[...] This provides direct empirical evidence [...]". The term 'direct evidence' should be used cautiously in this context. Causal inference tools, while valuable, have limitations in providing conclusive evidence without experimental manipulation --I see this point was already made by another reviewer, but it is critical to stress. The study would benefit from a more thorough discussion of experimental designs that could address these limitations, such as transmission chains or more complex microsocieties. These methods can indeed provide critical information on the mechanisms behind the origins and persistence of cultural variation(Mesoudi 2007), in all its forms (production, perception, consumption).

While the Limitations section briefly mentions these approaches, it lacks a precise description of how they could overcome the current methodological constraints. In this regard, a short section primarily focuses on mass online designs (e.g., Anglada-Tort et al. 2023), which, while offering large sample sizes and cross-cultural implementation, may lack the rigorous experimental control of laboratory designs. To strengthen this aspect, the authors should consider *only briefly* expanding on how laboratory methods could be applied specifically to their research questions, and how that could address current limits (e.g., manipulating critical variables, randomly assigning micro-societies with different demographical characteristics to different experimental and control conditions, etc.). Additionally, it would be beneficial to cite examples of such methods already implemented in the musical domain, such as Ravignani et al. 2016, Lumaca et al. 2017, Verhoef and Ravignani 2021, and Popescu and Rohrmeier 2022. These four examples

provide concrete illustrations of how mechanistic questions on the origins of musical diversity (and universals) have been addressed using more controlled experimental designs.

Thank you for these suggestions. We now tone down the argument on direct causal evidence:

***L384:** This provides supporting evidence for the ‘cultural breadth’ hypothesis that thus far remained largely theoretical.*

We also include references suggested by the reviewer, which are all highly relevant and expanded future experimental design section in the Limitations:

***L518:** Nevertheless, the advent of online experimental platforms and advanced recruitment methods now enables large-scale and cross-cultural studies. Particularly in the domain of music, iterative transmission paradigms have enabled the emergence of musical universals and diversities in controlled lab settings, providing insights into their mechanisms (78–82). Future experiments could, for instance, allocate participants randomly across different micro-societies (83) to produce or consume a specific cultural medium. These micro-societies can then be manipulated in their size, flow of immigrants (i.e., new participants), and the breadth of each participant’s interaction controlled by the network structure—as previously done with research in language (84). Such future experimental approaches for determining causality can go in tandem with our large observational analyses of real-world behaviour for answering questions about the general mechanisms underlying cultural evolution and transmission.*

R1.4: "Using causal inference tools, we found that demographic factors largely contribute to diversity in the most populated areas. However, they did not fully account for the observed trend of increasing diversity with population size. This potentially suggests that cultural diversity found in large metropolises is not only due to demographic mixing, but also results from more frequent cultural interactions and exchanges that these larger cities enable."

The discussion on the relationship between population size and cultural diversity could be more nuanced and comprehensive. While the authors touch on an important aspect - that larger populations facilitate more frequent interactions - this explanation feels somewhat rushed. The authors should consider expanding briefly this section to include additional mechanisms through which population size might influence cultural diversity. One example is global connectivity: Larger cities often serve as hubs in global networks, facilitating the introduction and integration of international cultural products into local traditions. This enhanced connectivity can significantly contribute to cultural diversity.

We agree that there can be additional mechanisms and they warrant a discussion. We now included the suggested perspective to the Discussion:

***L404:** This suggests that cultural diversity in large metropolises is not only due to demographic mixing but also results from additional mechanisms. One such mechanism may be the more frequent cultural interactions and exchanges that larger populations enable, promoting diversity through increased interpersonal contact. Another is the enhanced global connectivity of big cities, as international hubs, they facilitate the introduction and integration of global cultural products into local traditions.*

R1.5: "Cultural traits resulting from production and consumption interact intimately through feedback cycles, which are common in complex systems, including culture and economy (43,70,71). "

Consumption patterns may not only influence the range of what is produced but can also inspire the blending of genres or the emergence of entirely new forms (*innovation*). When consumers show interest in more diverse cultural products, producers may respond by experimenting with new combinations of styles and ideas to cater to that demand. This increased diversity in production patterns could then further enhance diversity in consumption and/or perception, reinforcing the feedback loop discussed in the manuscript.

Thank you for proposing this, we now added to the Discussion:

***L430:** Cultural traits resulting from production and consumption interact intimately through feedback cycles, which are common in complex systems, including culture and economy. For instance, a diverse range of consumption tastes might stimulate producers to create a broader array of cultural products, with social feedback signalling this demand. Conversely, individuals exposed to a wider range of cultural inputs, and consequently possessing a broader cultural breadth, may naturally tend to develop cultural niches. Consumption patterns may not only influence the range of what is produced but can also inspire innovations through blending of genres or the emergence of entirely new forms. When consumers show interest in more diverse cultural products, producers may respond by experimenting with new combinations of styles and ideas to meet that demand. This increased diversity in production patterns could then further enhance diversity in consumption, reinforcing the feedback loop.*

Reviewer 3

R3.1: Title: The new, shortened title does not quite make grammatical sense to me – I think you need to add the word ‘understanding’, ‘investigated’ or similar to the beginning.

We agree with the reviewer and propose a new title that is concise and potentially appeal to a broader audience:

“Why cities vary in their cultural diversity”

R3.2: Abstract: Should ‘250 million real-world listening behaviours’ read ‘250 million real-world listening events’?

We now corrected this:

L26: Leveraging data from over 2.5 million French, Brazilian, and German listeners, comprising 250 million real-world listening events

R3.3: There are some minor language and grammatical issues remaining throughout, e.g. ‘faster pace lifestyle’ should be ‘faster paced lifestyle’, ‘show to expand with population size’, ‘ominovorousness’ should be ‘omnivory’

We corrected the points raised by the reviewer and also took this opportunity to thoroughly proofread the manuscript for further grammatical issues. These changes are all highlighted in the revised manuscript.

L41: Inhabitants of large metropolitan areas tend to live a faster paced lifestyle

L69: Furthermore, while the link between socio-economic status and cultural breadth is well-studied (e.g., cultural omnivory)

R3.4: Is it possible to explain the units of BID in an intuitive way? The per-country means (last paragraph of p4) are challenging to interpret.

We appreciate the suggestion by the reviewer and now include how the values should be interpreted in the Figure 1 caption and Methods:

L160: The resulting BID value X indicates that the diversity is equivalent to having X cultural items (i.e., songs, artist, genre categories) that are all equally represented in the dataset.

L625: From the 1,000 music streams we sample in each bootstrap (Statistical analysis in Methods), a Hill’s number of 900, for instance, implies that the diversity is equivalent to

having 900 songs that are equally represented in the dataset—that is, how many equally common songs would produce the observed diversity?

R3.5: The way that ‘song embedding’ is explained is now less technical, but perhaps too abstract. It may be helpful to clarify that the ‘mathematical space’ is multidimensional and to explain what particular aspects of song style these dimensions capture (e.g. which rhythmic, harmonic or melodic etc. variables are included, end of p.17/start of p.18). Confusingly, in the next sentence ‘Deezer employs the singular value decomposition...’ it sounds like the measure of song similarity is actually based not directly on stylistic similarities but rather the co-occurrence of songs in user-generated playlists.

We are sorry for this confusion. As the reviewer notes, the embedding is a multidimensional space that capture the similarity between songs based on ‘co-occurrences of songs’, which often corresponds to musical genres and styles (e.g., songs that are same genre tends to also have high co-occurrence). We now understand that the notion of musical style might imply the acoustic aspects and be a source of confusion, we thus revised the section as follows:

***L591:** Embedding is a multidimensional space where objects, such as songs in this context, are represented as vectors in a way that captures the relationships between these objects based on user behaviour, namely co-occurrences of songs across playlists and listening patterns. In this space, closely related songs are positioned near each other. Tracks that share thematic or stylistic similarities tend to also have high co-occurrences. For instance, Adele's ‘Someone Like You’ and Sam Smith's ‘Stay With Me’ would be located close to each other within this space. To provide a tractable low dimensional space, Deezer employs the singular value decomposition (SVD) technique based on the co-occurrence of millions of songs.*

R3.6: When describing Hill’s number, it would be helpful to clarify that ‘assemblage’ = ‘species assemblage’ (p. 18)

We now corrected this:

***L617:** ... has become an increasingly popular method to quantify the diversity of a species assemblage.*

R3.7: Please mention and cite and specific R packages that were used in the analyses (p. 24)

We now added missing package citation and mention how all other general stats were computed:

***L812:** To control for confounders, we used propensity scores to adjust for group differences in users living in different size areas using the ‘WeightIt’ R package (99)*

L832: *Analysis was conducted using R (version = 4.3.3). Unless explicitly mentioned, all stats were computed using the 'stats' package in base R and custom scripts (Code availability).*

Response to the Reviewers

Apr 25, 2025

Dear Reviewer 3,

Thank you for the attention to detail in spotting grammatical errors and providing further constructive suggestions for the paper entitled "*Why cities vary in their cultural diversity*".

We have fully addressed all the issues raised and revised the manuscript as provided in a detailed response below.

To facilitate an easier task for the reviewers, we labelled each point raised by the reviewers and their comments as **bold** text. Our response is placed just under as normal text and reference to changes in the revised manuscript as *italic* text including line numbers in brackets. In the revised manuscript, all changes are **highlighted in yellow**. In addition, we provide a clean version with all changes resolved.

Sincerely,

Harin Lee, Nori Jacoby, Romain Hennequin, Manuel Moussallam

Reviewer 3

Line 4 of the abstract might be clearer if phrased: 'could not discern how individual factors contribute to collective cultural outcomes'

Edited text: '*...previous research could not discern how individual factors contribute to collective cultural outcomes.*'

Line 7 of the first paragraph of the introduction: remove 'show to'

Edited text after removal: '*Cultural richness and complexity also expand with population size*'

Line 1 of page 3: I think this should read 'with underlying selection pressures', and you can remove 'the' before 'cultural evolutionary processes'

Edited text: '*...feedback loops with underlying selection pressures, are crucial for a comprehensive understanding of cultural evolutionary processes.*'

Line 4, second paragraph of p10: should read 'and the greatest availability of...'?

Edited text: '*...spatial granularity there and the greatest availability of dense...*'

Fig 3 legend: here a little further clarifications would be appreciated by non-specialist readers, including what is meant by an 'exposure' variable and what the thick black arrows in panels a & b indicate

Thanks for this suggestion, we now updated the caption as follows:

'DAG models for testing the direct effects of population size (exposure; i.e., the factor being tested) on (a) BID and (b) WID (outcome; i.e., the response being measured). Adjusted variables in the models (i.e., minimal adjustment set) are coloured in red, whereas unadjusted variables are coloured in grey (Causal inference in Method). Bold arrows indicate the causal paths of interest.'

Line 6, second paragraph of p. 14: perhaps remove the word 'naturally' as this might be misinterpreted as meaning 'innate' or similar.

Edited text: '*...a broader cultural breadth, may tend to develop cultural niches...*'